# CEP44 ensures the formation of bona fide centriole wall, a requirement for the centriole-to-centrosome conversion

Enrico S. Atorino [1], Shoji Hata [1], Charlotta Funaya[2], Annett Neuner[1] & Elmar Schiebel [1✉]

Centrosomes are essential organelles with functions in microtubule organization that duplicate once per cell cycle. The first step of centrosome duplication is the daughter centriole formation followed by the pericentriolar material recruitment to this centriole. This maturation step was termed centriole-to-centrosome conversion. It was proposed that CEP295-dependent recruitment of pericentriolar proteins drives centriole conversion. Here we show, based on the analysis of proteins that promote centriole biogenesis, that the developing centriole structure helps drive centriole conversion. Depletion of the luminal centriole protein CEP44 that binds to the A-microtubules and interacts with POC1B affecting centriole structure and centriole conversion, despite CEP295 binding to centrioles. Impairment of POC1B, TUBE1 or TUBD1, which disturbs integrity of centriole microtubules, also prevents centriole-to-centrosome conversion. We propose that the CEP295, CEP44, POC1B, TUBE1 and TUBD1 centriole biogenesis pathway that functions in the centriole lumen and on the cytoplasmic side is essential for the centriole-to-centrosome conversion.

[1] Zentrum für Molekulare Biologie, Universität Heidelberg, DKFZ-ZMBH Allianz, 69120 Heidelberg, Germany. [2] Electron Microscopy Core Facility, Universität Heidelberg, 69120 Heidelberg, Germany. ✉email: e.schiebel@zmbh.uni-heidelberg.de

The centrosome is the main microtubule-organising centre (MTOC) of higher eukaryotic cells[1,2]. This organelle comprises a central cylindrical tubulin-based structure, the centriole[3], and a protein matrix that surrounds the centriole called the pericentriolar matrix (PCM)[4–6]. The PCM is a highly organised toroid matrix in which the proteins that constitute it are concentrically placed in it and emanate from the proximal end region of the centriole[5]. One of the main functions of the PCM is the microtubule (MT) nucleation[7–9]. The centriole is a cylindrical structure made up of MT-triplets arranged in a 9-fold radial symmetry[10,11]. Each triplet consists of an inner tubule of 13 tubulin protofilaments (the A tubule) to which two additional tubules, the B and then the C tubules, are attached[12].

Centrosomes duplicate once per cell cycle. The newly born daughter centriole (dC) grows perpendicularly on the mother centriole (mC) and is kept engaged on it until the cell enters the next cell cycle[13]. During the process of the dC elongation, its structure grows radially and builds MT triplets via TUBE1 and TUBD1[14].

After centriole formation in S/G2 phase, the dC is converted into a centrosome by the step-wise recruitment of PCM proteins. This process called centriole-to-centrosome conversion (CCC)[15,16] is essential for the new centrosome to gain MT nucleation activity and the ability to duplicate. The current model, derived from *D. melanogaster* and different human cultured cell lines, suggests a PCM protein recruitment cascade on the dC wall. The centriole proximal end protein CEP135 binds to the centriole wall and connects to CEP295, a key protein in CCC. CEP295 recruits the PLK4 kinase adaptor CEP152 and the γ-tubulin recruitment factor CEP192 to the new dC[16,17]. In human cells CCC was described to stabilise the centrioles in mitosis[15]. Therefore, loss of CCC impairs centrosome homeostasis[18] and causes developmental defects, e.g. microcephaly, ciliopathies and cancer[19–21].

How CCC works on a molecular level is little understood. Interestingly, the processes of centriole maturation and CCC occur simultaneously raising the possibility of interplay between the unique 9-fold MT-triplet structure and PCM recruitment to the dC. To better understand the CCC mechanisms we focus on centrosome biogenesis studying CEP44, an essential centrosomal protein[22,23]. Our analysis identifies CEP44 as a luminal centriolar protein that binds to A-MTs and the inner centriole protein POC1B. CEP44 depletion impairs CCC although the binding of CEP295 to the dC is unaffected indicating additional CCC mechanisms than the CEP295-dependent recruitment of PCM proteins. Interestingly, also depletion of POC1B, TUBE1 and TUBD1 affects CCC beside centriole structure. We therefore propose that centriole biogenesis via CEP295, CEP44, POC1B, TUBE1 and TUBD1 pathway is important for the conversion of centrioles to fully-functional centrosomes.

## Results

### CEP44 is a centriole protein necessary for the CCC mechanism.
Our interest in understanding the biogenesis and the function of the centrosome drove us to characterise CEP44 out of novel centrosomal proteins[22] because it was the only of these that was described to be essential in human cells[23]. Affinity purified antibodies rose against the protein showed that in G1 and S phase (EdU positive) cells, CEP44 localised to mCs whereas in G2 cells it appeared as 4 foci localising to all 4 centrin1 signals, the two mCs and two dCs (Fig. 1a). This led us to conclude that CEP44 is a centriolar protein.

In order to understand the function of CEP44 at centrioles, we knocked down the expression of CEP44 in RPE1 h-TERT cells via siRNA for 72 h (Fig. 1b and Supplementary Fig. 1a–d) and

quantified the number of centrosomes counting the γ-tubulin foci (Fig. 1c). Only G1 cells (negative for EdU and Cenp-F[24]) (Supplementary Fig. 1b for Cenp-F) were considered in this quantification in order to simplify the analysis. In the 72 h CEP44 knockdown sample, $80.4 \pm 5.0\%$ of G1 cells displayed less than two defined γ-tubulin signals (Fig. 1b, c). Analysis of the presence of centrioles in CEP44 depleted G1 cells judging the number of centrin1 foci, showed the loss of centrioles in $35.0 \pm 2.8\%$ of cells (Supplementary Fig. 1c, d).

The clear stronger impact on γ-tubulin than on centrin1 suggested that the PCM recruitment defect arises before the centriole loss. We explored this possibility further at an earlier time point of CEP44 depletion (60 h) when the centriole loss in G1 was minimal ($5.1 \pm 1.6$, Supplementary Fig. 1e, f). 60% of the cells still showed a strong defect in the recruitment of both PCM components γ-tubulin and PCNT, which corresponded in its magnitude with the CEP44 loss (Fig. 1d–f; Supplementary Fig. 1e–i). Moreover, the PCM recruitment defect associated with the centrioles lacking the mC marker CEP164[25], thus with the dCs (Supplementary Fig. 1j–l).

We explored whether the loss of the centrioles in G1 after 72 h depletion (Fig. 1b) was due to a centriole duplication defect or instability of the centrioles. Therefore we tested the duplication of centriole in G2 cells at 60 h upon CEP44 depletion by counting the number of the foci of the centriolar marker CEP97[26] (4 or fewer). We observed that $30.3 \pm 4.1\%$ of G2 cells in the siCEP44 sample showed <4 CEP97 foci, thus a defect in the ability of the centrioles to duplicate (Fig. 1g and h), explaining the loss of centrioles in the subsequent G1 phase at 72 h (Supplementary Fig. 1c and d). In all following experiments we depleted CEP44 for 60 h and analysed G1 cells in order to avoid a centriole duplication defect. This together indicates that CEP44 primarily plays a role in the CCC of the dC and not in the maintenance of mother centrosomes.

We confirmed that CEP44 was responsible for the CCC phenotype. We rescued the defect of dCs to recruit PCM by expressing an exogenous siRNA-resistant *CEP44* version, which completely restored centrosome function upon siCEP44 treatment of the cells (Supplementary Fig. 1m, n). In addition, we analysed *CEP44* knockout (KO) cells that were constructed by a CRISPR/Cas9 approach[27]. Bi-allelic *CEP44* KOs were only obtained in a RPE1 *p53*$^{-/-}$ background cell line (Supplementary Fig. 2a, b), which is consistent with a function of CEP44 in CCC[23]. The KO cell lines showed the same PCM recruitment phenotype as described for the CEP44 siRNA-induced knockdown (Supplementary Fig. 2c–e). Similarly to the knockdown rescue data, we were able to restore proper converted centrosome number by expressing exogenously *CEP44* in *CEP44*$^{-/-}$ *p53*$^{-/-}$ cells (Supplementary Fig. 2c–e).

### CEP44 depletion leads to MT nucleation and mitotic defects.
A defect in γ-tubulin recruitment should affect the MTOC activity of centrosomes. RPE1 *C-Nap1* (KO) cells[28] were exploited for this defect because their non-cohered centrosomes allowed us to discern the MTs generated from the mother and the daughter centrosomes and thus to quantify them. CEP44-depleted daughter centrosomes displayed reduced MT nucleation activity in a MT regrowth assay compared to control daughter centrosomes (Fig. 1i, j) consistent with the failure of dCs to recruit γ-tubulin and PCNT (Fig. 1f and Supplementary Fig. 1h). As a consequence, CEP44 depletion severely compromised the ability of cells to assemble symmetrically balanced mitotic spindle poles (Fig. 1k, l). We conclude that deficiency in CEP44 recruitment to centrosomes perturbs the homeostasis of centrioles and the organisation of mitotic spindle poles.

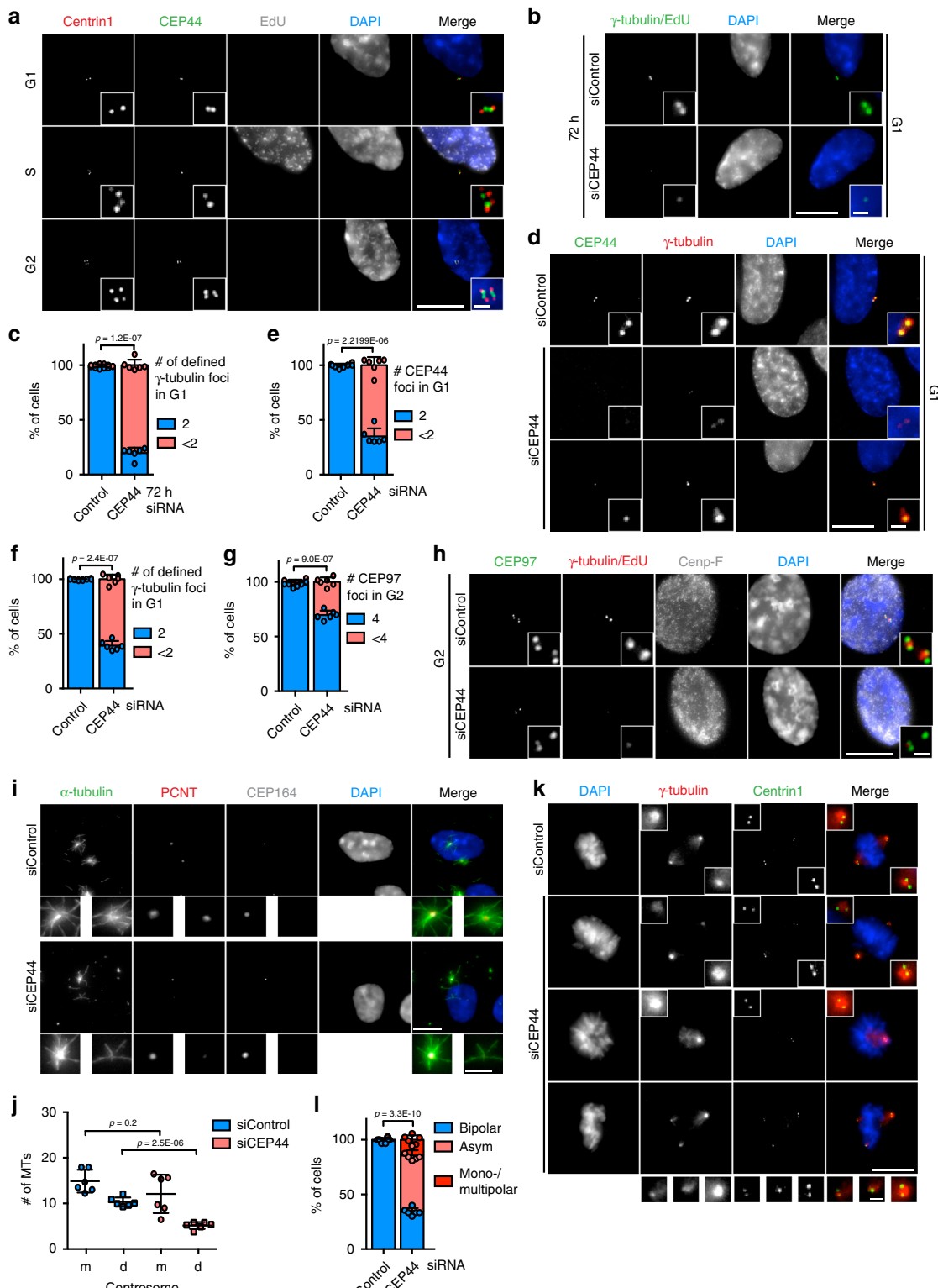

**CEP44 influences the CCC mechanism downstream of CEP295.** CEP44 depletion impaired PCM recruitment to the dC suggesting a function in CCC (Fig. 1). CEP295 was previously described to be the master regulator of the CCC[15–17]. We therefore expected that CEP44 loss would affect CEP295 recruitment to dCs. Surprisingly, G1 CEP44-less centrioles that were not converted still contained 2 CEP295 foci similarly to converted centrosomes in control cells (Fig. 2a, b, Supplementary Fig. 3a, b). On the other hand, CEP295 loss did affect both the ability of the dCs to convert

to centrosome and the localisation of CEP44 in G1 cells (Fig. 2c, d, Supplementary Figs. 1a and 3c–e). These experiments are consistent with a linear CCC model in which CEP44 functions downstream of CEP295. To confirm that CEP295 and CEP44 function in the same pathway, we compared the CCC phenotype of cells with single and double depletion of CEP44 and CEP295. Depletion of CEP295 and CEP44 did not increase the severity of the CCC defect in comparison to the single depletions (Supplementary Fig. 3f–h).

**Fig. 1 CEP44 is a centriolar protein necessary for the CCC mechanism. a** IF of cycling RPE1 cells showing that CEP44 binds to the new dCs during G2. While S phase cells were detected by EdU stain, G1 and G2 cells were discerned by the lack of EdU stain and the number of centrin1 signals (centrioles). **b** IF of cells after 72 h of depletion. In the siCEP44 sample (lower panel) G1 cells contained less centrosomes as judged by the number of γ-tubulin foci (Cenp-F in Supplementary Fig. 1b). **c** Quantification of **b**. 80.4 ± 5.0% of G1 cells contained <2 centrosomes. **d** IF of 60 h siRNA treated RPE1 cells in G1. While the G1 control cells contained 2 defined PCM foci (γ-tubulin) accompanied by equal number of CEP44 foci, in the siCEP44 sample the loss of CEP44 correlated with inefficient PCM (γ-tubulin) recruitment to only one (bottom panel) or both centrosomes (middle panel) (Cenp-F, Supplementary Fig. 1g). **e** Quantification of CEP44 loss in **d**. 65.1 ± 7.3% of G1 cells contained <2 CEP44 foci. **f** Quantification of γ-tubulin defined foci in **d**. 60.7 ± 4.2% of G1 cells contained <2 defined γ-tubulin signals. **g**, **h** G2 CEP44-depleted cells after 60 h of CEP44 depletion showed a mild centriole duplication defect as judged by the counting of CEP97 foci (<4). **g** Quantification of **h**. 30.3 ± 4.1% of CEP44-depleted cells contained <4 centrioles. **i** MT regrowth assay in RPE1 *C-Nap1* KO cells. The non-converted daughter centrosome without CEP164 staining regrew lower numbers of MTs (5.1 ± 2.7 MTs/centrosome) than the siControl daughter centrosomes (10.3 ± 3.0 MTs/centrosome) upon cold treatment and MT regrowth. **j** Quantification of **i**. **k** Loss of CEP44 leads to misalignment of mitotic spindles (>65%) generating either bipolar asymmetric spindles (51.3 ± 4.7%) or mono-/multipolar ones (14.3 ± 4.0). **l** Quantification of **k**. (**a**, **b**, **d**, **h**, **i**, **k**, scale bars: 10 μm, magnification scale bars: 1 μm; **c**, **e**, **f**, **g**, **j** and **l** data are presented as mean ± s.d., all statistics were derived from two-tail unpaired *t*-test analysis of *n* = 6 biologically independent experiments and source data are provided as a Source Data file).

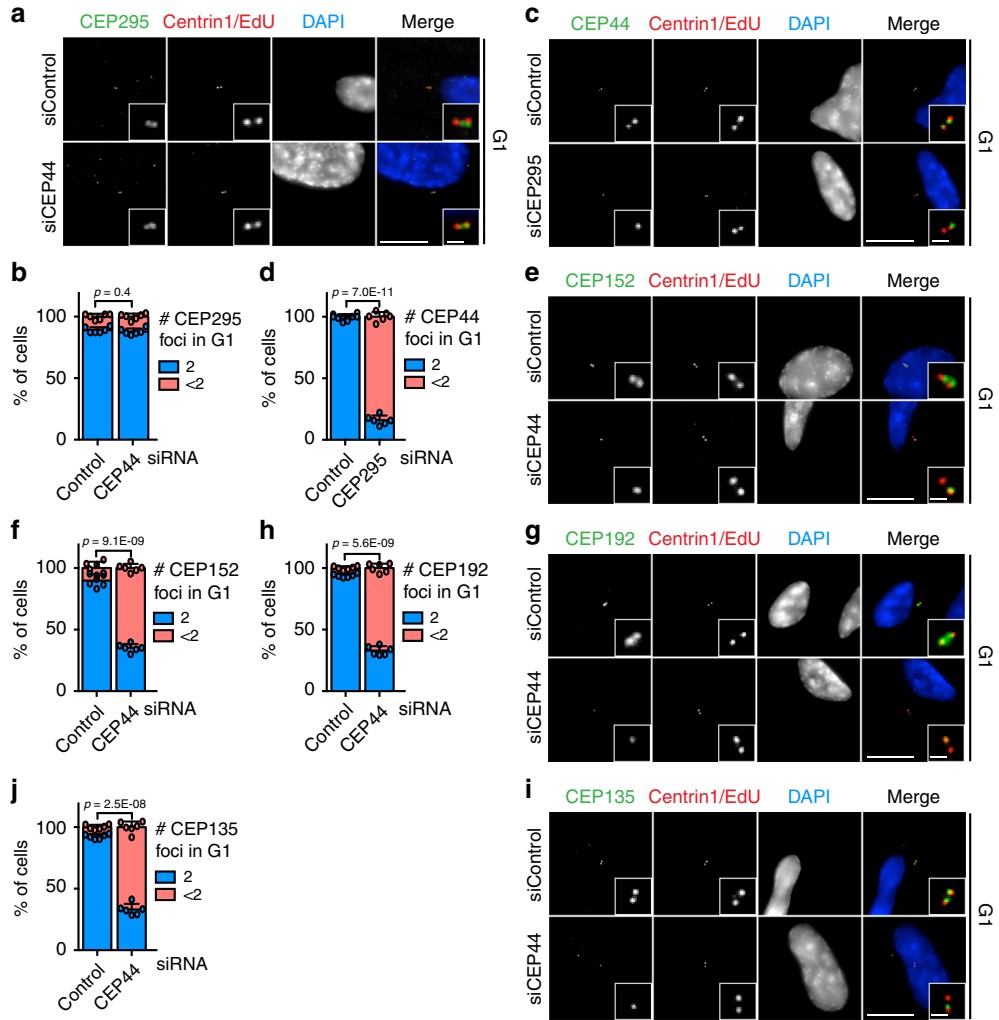

**Fig. 2 CEP44, downstream of CEP295, influences the conversion mechanism. a** CEP295 did not delocalise from the centrioles in G1 cells (Cenp-F in Supplementary Fig. 3a) upon CEP44 siRNA depletion (10.9 ± 2.3% of cells with <2 CEP295 foci in siControl cells vs. 12.1 ± 2.5% in siCEP44 sample). **b** Quantification of **a**. **c** CEP44 strongly delocalised from the daughter centrosome in the absence of CEP295. G1 cells (Cenp-F negative in Supplementary Fig. 3c) were analysed. **d** Quantification of **c**. In siCEP295 84.0 ± 3.7% of the cells had <2 CEP44 foci. **e** CEP44-less daughter centrosomes lacked CEP152 (**f**, 64.9 ± 3.2% of G1 cells vs. 10.1 ± 5.1% in control; Cenp-F in Supplementary Fig. 3i, left column), **g** CEP192 (**h**, 67.1 ± 3.6% of G1 cells; Cenp-F in Supplementary Fig. 3i, middle column) and **i** CEP135 (**j**, 66.9 ± 4.6% of G1 cells; Cenp-F in Supplementary Fig. 3i, right column). (**a**, **c**, **e**, **g**, **i**, scale bars: 10 μm, magnification scale bars: 1 μm; **b**, **d**, **f**, **h** and **j** data are presented as mean ± s.d., all statistics were derived from two-tail unpaired *t*-test analysis of *n* = 6 biologically independent experiments and source data are provided as a Source Data file).

We next tested whether CEP44 is necessary for the recruitment of CEP152 and CEP192 that bridge the centriole to the PCM[16]. CEP152 and CEP192 localised to centrioles in the G1 control cells, while in CEP44 siRNA-treated cells they were delocalised (Fig. 2e–h, Supplementary Fig. 3i–k). In particular, the loss of the ability of the dCs to recruit CEP152 efficiently (Fig. 2e, f, Supplementary Fig. 3j), an adaptor for PLK4 kinase that promotes dC formation[29,30], explains the duplication defect in response to CEP44 depletion (Fig. 1g, h).

In RPE1 cells CEP135 was delocalised on dCs upon CEP44 depletion and thus functions downstream of it (Fig. 2i, j, Supplementary Fig. 4a). In *D. melanogaster* CEP135 was reported to be important for the recruitment of CEP295[16]. This finding in *D. melanogaster* is inconsistent with the binding of CEP295 to CEP44-less dCs in RPE1 cells that fail to recruit CEP135. We therefore tested the correlation between the proteins and the contribution of CEP135 to the CCC. Analysis of the cell-cycle recruitment in RPE1 cells unveiled that CEP295 is recruited to the new dC earlier than CEP135 (Supplementary Fig. 4b–e) and thus temporally upstream CEP135. Moreover, depletion of CEP135 did not affect the recruitment of γ-tubulin to the centrioles in G1 cells (Supplementary Figs. 1a and 4f, g).

From these data we conclude that CEP44 in respect of dC recruitment acts upstream of CEP135, CEP152 and CEP192 but downstream of CEP295. In addition, CEP295 is essential but insufficient for the recruitment of PCM to the new dCs.

**CEP44 interacts with POC1B**. To better understand the role of CEP44 in the CCC, we analysed the ability of CEP44 to interact with centrosomal proteins. Mass spectrometry analysis of proteins immunoprecipitated (IP) with CEP44-Flag identified POC1B and POC1A as CEP44 interactors (Supplementary Table 1). Other proteins with known functions in CCC or centriole duplication were not detected in this experiment. A CEP44-Flag IP, performed with a higher salt concentration than the IP mass spectrometry experiment (see Methods), confirmed the interaction of CEP44 with POC1B (Fig. 3a). Under this condition POC1A was not detected in the CEP44 IP suggesting that the CEP44-POC1A interaction is more labile than that of CEP44-POC1B. Consistent with the mass spectrometry analysis, CEP295 was not detected in this CEP44-Flag IP (Fig. 3a). The interaction between CEP44 and POC1B was further confirmed using *E.coli* purified recombinant proteins (Fig. 3b, Supplementary Fig. 5b, c). Further support for an interaction between CEP44 and POC1B came from the ability of overexpressed CEP44 to shift the localisation of POC1B from the cytosol to MTs (Supplementary Fig, 5d, e). In contrast, the N-terminal (NT) half of CEP44 (Fig. 3d, aa 1-195) was unable to recruit POC1B to MTs (Supplementary Fig. 5d, f).

To test, which region of CEP44 interacts with POC1B, we analysed CEP44 for subdomains. GeneTree alignment (ENSGT00390000009873) and protein sequence alignment indicated that CEP44 in vertebrates is highly conserved with an amino acid (aa) identity of >60% (Fig. 3c, Supplementary Fig. 5a) at the N-terminal half of the protein. This 130 aa N-terminal region contains the already-annotated CEP44 domain[31]. In contrast, the CEP44 C-terminus is more divergent (Fig. 3c). Based on this analysis we designed CEP44 constructs to determine the region that interacts with POC1B (Fig. 3d). Analysis of the CEP44-Flag immunoprecipitations showed that almost the full-length CEP44 protein (aa 1–310) with the exception of the C-terminal 80 aa was required for the binding to the endogenous POC1B (Fig. 3e). Thus, CEP44 directly interacts with POC1B.

**CEP44-POC1B interact and ensure fully dCs conversion**. Being an interactor of CEP44, we tested whether POC1B is also necessary for CCC. Interestingly, POC1B siRNA treatment of RPE1 cells showed that, similarly to CEP44-loss, G1 centrioles missing POC1B failed to recruit the PCM proteins γ-tubulin and CEP192 (Fig. 3f, g, Supplementary Fig. 5g for Cenp-F and Supplementary Fig. 5h, i). The centriolar POC1B siRNA depletion efficiency was similar to the inability to recruit γ-tubulin to centrioles (Supplementary Figs. 1a and 5j).

We next tested the hierarchical relationship between CEP44 and POC1B. G1 cells with CEP44 depletion carried centrioles that lacked POC1B (Fig. 3h, i, Supplementary Fig. 6a, b). The loss of POC1B upon CEP44 depletion was fully rescued by over-expressing a siRNA CEP44-resistant recombinant protein, but only mildly by the NT-CEP44 fragment (Supplementary Fig. 6a, b). On the other hand, POC1B depletion only mildly affected the recruitment of CEP44 to dCs, less than its depletion efficiency (Fig. 3h, j). This localisation analysis positions POC1B downstream of CEP44. However, POC1B has a mild impact on the localisation of CEP44, which is consistent with the interaction of both proteins (Fig. 3k).

We next analysed a role of the POC1B paralogue POC1A in CCC. siPOC1A that efficiently reduced POC1A at centrosomes (Supplementary Fig. 7a) barely affected CCC (Supplementary Fig. 7b, c). However, it is still possible that a role of POC1A in CCC is masked by the compensatory presence of POC1B. This was tested by POC1A and POC1B double depletion experiments (siPOC1A + B). As reported before[32], siPOC1A + B impaired centriole duplication as indicated by the reduced number of centrin1 signals in G1 cells (Supplementary Fig. 7d, e). In addition, CEP295 and CEP44 localisation with centrioles was more strongly affected by siPOC1A + B than by the depletion of only POC1B (Supplementary Fig. 7g–k, Fig. 3i). However, CEP44 depletion still delocalised POC1B stronger (59.6 ± 2.7%) than siPOC1A + B impacted CEP44 (29.5 ± 3.0%) (Fig. 3i, Supplementary Fig. 7h). siPOC1A + B affected the localisation of γ-tubulin to duplicated centrioles stronger (58.2 ± 12.8%) than the single depletion of POC1A (9.6 ± 2.0%) or POC1B (39.2 ± 2.8%) (Fig. 3g, Supplementary Fig. 7c, f). These phenotypes indicate that loss of both POC1A and POC1B has a stronger impact on centriole biogenesis than single depletion of POC1A or POC1B[32].

**CEP44 and POC1B localise to the lumen of centrioles**. The function of CEP44 downstream of CEP295 and the CEP44-POC1B interaction raised the question of their layout with sub-domains of centrioles (Fig. 3k). To achieve sufficient optical resolution, we analysed the localisation of CEP295, CEP44 and POC1B by 2D-Structured Illumination Microscopy (2D-SIM). The CEP44 signal at mCs was restricted to the lumen of centrioles (the wall was marked by antibodies against α-tubulin) (Fig. 4a, Supplementary Fig. 8a). POC1B similarly to CEP44 was also localising to the lumen of the centriole (Fig. 4b, Supplementary Fig. 8b). Differently from these two CCC molecules, CEP295 decorated the outer side of the mCs wall (Fig. 4c).

Because these proteins are important for the conversion of the dC, we extended our analysis on their localisation to the newborn dCs. We plotted the intensity profiles of cross sections of centrioles that grew perpendicularly to pre-existing ones, thus dCs. CEP44 and POC1B both localise inside the new dC structure (α-tubulin) (Fig. 4d, e). The cross section plot profile of CEP295 showed that the protein localised to the outer side of the centriole wall of dCs (Fig. 4f). CEP44 localised in both mCs and dCs to the proximal end of the centriole (Fig. 4d and Supplementary Fig. 8c). In mCs POC1B partially localised to the proximal region, but mostly to the middle-distal end (Fig. 4d and

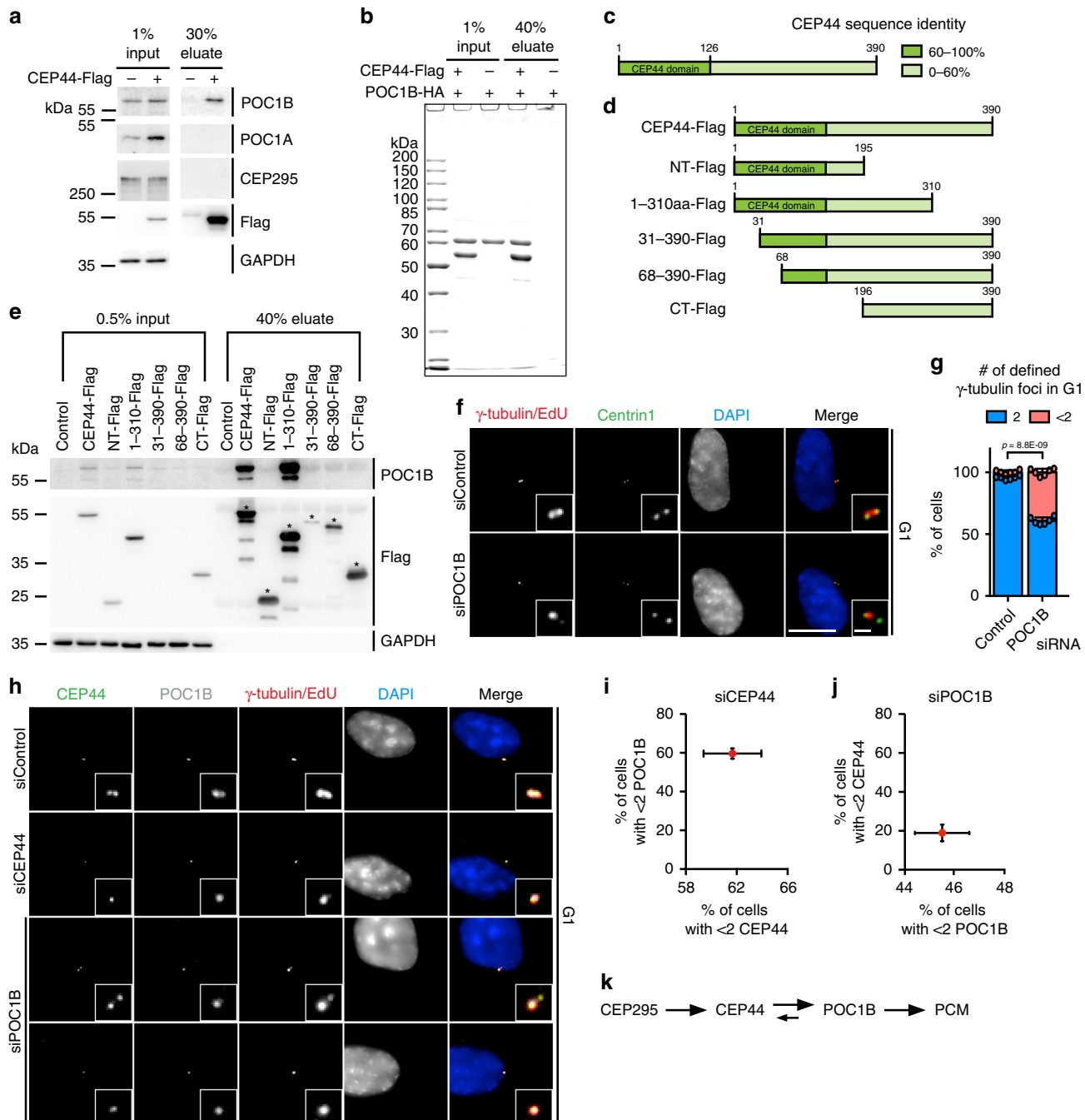

**Fig. 3 The CEP44-POC1B complex is needed to convert centrioles to centrosomes. a** Anti-Flag IP using CEP44-Flag from RPE1 cells was analysed for POC1B, POC1A and CEP295 by immuno-blotting (IB). GAPDH was used as input control. **b** Coomassie Blue stained gel of in vitro binding between purified, recombinant CEP44-Flag and purified, recombinant POC1B-HA. See Supplementary Fig. 5b for IB and 5c for Coomassie Blue stained gels of purified proteins used in the experiment. **c** Schematic representation of CEP44 protein sequence identity in vertebrata (referred to Supplementary Fig. 5a). **d** CEP44-Flag constructs that were designed based on **c** and used in **e**. **e** IB of input and eluted samples from RPE1 IPs using CEP44-Flag constructs as outlined in **d**. The CEP44-Flag IPs were tested for the presence of POC1B. GAPDH was used as loading control for the input. **f, g** 39.2 ± 2.8% of G1 cells in which POC1B was depleted show <2 γ-tubulin defined foci (Cenp-F in Supplementary Fig. 5g). **h–j** Loss of either CEP44 or POC1B in response to siRNAs depletion by one of them. **h, i** CEP44 loss upon CEP44 siRNA has a similar impact on POC1B loss from dCs. **i** Quantification of **h**. **h, j** CEP44 delocalisation was less severe than POC1B loss upon POC1B siRNA. **j** Quantification of **h**. Upon siPOC1B depletion, CEP44 delocalised (18.9 ± 4.3% of G1 cells) less sever than POC1B (45.5 ± 2.3%). **k** Schematic representation of the functional interdependency between the conversion molecules in the CCC mechanism. (**f**, **h**, scale bars: 10 μm, magnification scale bars: 1 μm; **g**, **i**, **j** data are presented as mean ± s.d., all statistics were derived from two-tail unpaired *t*-test analysis of *n* = 6 biologically independent experiments and source data are provided as a Source Data file).

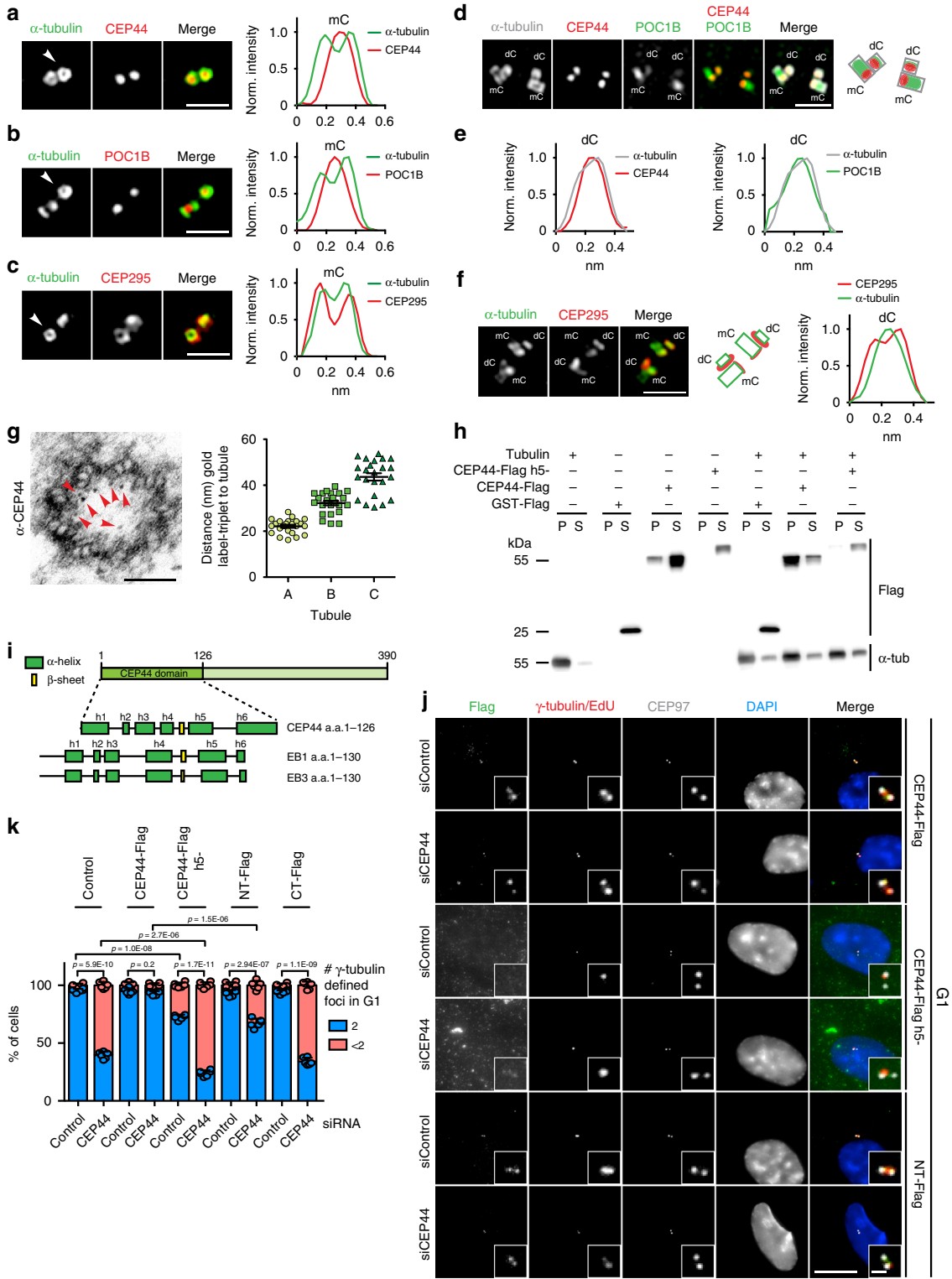

Supplementary Fig. 8c). Therefore, in mCs the profiles of CEP44 and POC1B intensities only partially overlapped. Most interestingly, however, in newly formed dCs CEP44 and POC1B signals completely overlap (Fig. 4d, Supplementary Fig. 8d).

To increase the resolution of CEP44 localisation in the proximal end lumen of the centriole, we performed immunogold labelling of CEP44 in purified centrosomes to determine its sub-localisation. Immuno-gold labels showed the localisation of CEP44 close to the MT-wall from the inner side (Fig. 4g, Supplementary Fig. 8e). Measuring the distances between the

label and either A-, B- or C-tubules, we found out that CEP44 localises close to the A-tubules (Fig. 4g, right panel). In conclusion, CEP295 and CEP44 localise on opposite sides of the centriolar MT wall, while POC1B co-localise with CEP44 in the dCs lumen and partially overlaps with it in the mCs.

**CEP44 MT-binding domain recruits a pool of POC1B to dCs.** The observation that CEP44 associates close with the A-tubules of centrioles promoted us to test whether it has the ability to bind to

**Fig. 4 CEP44 localises to the centriole lumen via its MT-binding affinity. a–c** 2D-SIM images of G1 centrioles (α-tubulin) and corresponding normalised intensity profiles of centrosomes positioned perpendicularly to the imaging plane (white arrows). **a** CEP44 localised in the centriole lumen as POC1B (**b**). **c** CEP295 decorates the outer centriolar wall. **d** 2D-SIM of centrosomes with duplicated centriole pairs co-stained with α-tubulin, CEP44 and POC1B. **e** Intensity profiles of dC cross-section. CEP44 (left) and POC1B (right) reside in the dC lumen. **f** (left) 2D-SIM of centrosomes with duplicated centriole pairs co-stained with α-tubulin and CEP295. **f** (right) Intensity profiles of dC cross-section. **g** CEP44 immuno-gold labelling in purified centrosomes (left). Red arrows indicate 10 nm gold particles. **g** (right) Distance of the gold particles from A-, B- and C-tubule of the same triplet respectively 21.9 ± 3.1 nm, 32.0 ± 4.7 nm, 43.1 ± 7.5 nm (all cases, $n = 23$ particles, data present mean ± s.d.). **h** Binding assay of recombinant CEP44-Flag and h5⁻ mutant purified from *E.coli* to MTs. GST-Flag was used as control. Proteins were incubated with soluble polymerised tubulin. MTs with bound proteins were sedimented by centrifugation. The supernatant (S) and pellet (P) were analysed by IB for α-tubulin and Flag. Supplementary Fig. 8f shows Coomassie blue stain gel of purified proteins. **i** Schematic representation of CEP44 domain organisation. (Bottom) Comparison of CEP44 domain predicted secondary structure organisation with the MT-binding domain of EB1 and EB3 proteins. **j** The h5⁻ and the NT-fragment could not rescue the CCC defect vs. CEP44-Flag. Constructs were mildly expressed by the addition of 2 ng/ml doxycycline. **k** Quantification of j and Supplementary Fig. 9c. While the CT-Flag was unable to rescue the loss of PCM (63.9 ± 3.2% of cells with <2 γ-tubulin foci) and the NT only partially (32.6 ± 3.8%), the h5⁻ mutant generated a CCC defect even in the siControl (27.6 ± 2.3%) and a stronger CCC phenotype in the siCEP44 (76.1 ± 2.6%). Data presented as mean ± s.d., all statistics derived from two-tail unpaired *t*-test analysis of $n = 6$ biologically independent experiments. (**a**, **b**, **c**, **d**, **f**, scale bars: 1 µm; **g**, scale bar: 100 nm; **j**, scale bar 10 µm, magnification scale bar: 1 µm). (**a–c**, **e–g**, **k**) Source data are provided as a Source Data file.

MTs. In vitro data showed that recombinant, purified CEP44 bound to taxol-stabilised MTs (Fig. 4h and Supplementary Fig. 8f). This affinity to MTs was confirmed by the localisation of overexpressed CEP44, which uniformly decorated all MTs of an interphase cell (Supplementary Fig. 8g, upper panel). Localisation of the overexpressed NT- or CT-half constructs showed that the MT-binding domain of CEP44 resided in the NT-half of the protein (Supplementary Fig. 8g, two lower panels). Indeed, the sequence of the NT-half of the protein is highly conserved on the aa level not only in vertebrates but also in unicellular organisms (Supplementary Fig. 8h) and its predicted secondary structure organisation resembles the one from EB1 and EB3, two MT plus end tip binding proteins[33] (Fig. 4i). We then muta-genized the *CEP44* sequence to disrupt the MT-binding affinity according to chemical-physical features found to be relevant in EB1 and EB3 to bind tubulin[33] (Supplementary Fig. 9a). Recombinant CEP44 h5− did not bind to polymerised MTs in vitro (Fig. 4h and Supplementary Fig. 8f). CEP44 h5−also failed to bind to centrioles upon overexpression in vivo (Fig. 4j, two middle panels, Supplementary Fig. 9b). Similarly, the CT-Flag CEP44 construct failed to bind to centrioles (Supplementary Fig. 9c, two lowest panels). Furthermore, *CEP44* MT binding mutants were no longer able to rescue the CCC defect upon CEP44 depletion. Instead, expression of *CEP44 h5⁻* in control cells caused both a dominant negative dC conversion phenotype (mutant *h5⁻*, Fig. 4j, k, Supplementary Fig. 9c) and the deloca-lisation of POC1B from dCs (Supplementary Fig. 9d, e), partially mimicking depletion of CEP44.

Interestingly, the overexpressed NT fragment of CEP44 (aa 1-195), which binds MTs and centrioles (Fig. 4h, Supplementary Fig. 9c, rows 7 and 8) but not POC1B (Fig. 3e), partially compensated for the CCC and POC1B localisation defect of CEP44 depletion (Fig. 4j, k, Supplementary Fig. 6b) despite its inability to interact with it (Fig. 3e). It is therefore plausible that NT-CEP44, for example by having an impact on the centriole structure, directs some POC1B to centrioles. However, complete recruitment of POC1B and CCC depended on full-length CEP44, thus the ability of CEP44 to interact with POC1B (Fig. 4k, Supplementary Fig. 6b).

**CEP295 stabilises new dCs.** A newly-born dC is a stable struc-ture as long as it retains the SASS-6[10] cartwheel. With removal of the cartwheel in mitosis CEP295 deficient centrioles become unstable[15]. Presently, it is still unclear whether CEP295 alone or only in combination with its CCC function stabilises the dC. CEP44 depletion generated the condition in which G1 centrioles lose PCM but not CEP295 (Fig. 2a). We therefore could compare

the stability of the centrioles in CEP44 or CEP295 siRNA treated cells, the one having CEP295 but no PCM and the other lacking both CEP295 and PCM (Supplementary Fig. 10a). Upon CEP44 or CEP295 depletion the cells were cold-treated in order to destabilise centrioles. G1 centrioles in CEP295-less cells were unstable in comparison to CEP44-less ones (Fig. 5a, b, Supple-mentary Fig. 10b, c for Cenp-F).

In *CEP44* RPE1 WT cells, the SASS-6 cartwheel is removed in mitosis. SASS-6 removal in CEP295 depleted cells triggers centriole instability[15]. To exclude that CEP44 daughter centro-somes retained the SASS-6 cartwheel structure and were therefore stable, we depleted CEP44 and checked if G1 centrioles went trough the cartwheel removal process (Fig. 5c, Supplementary Fig. 10c for Cenp-F). All control, CEP295-less and CEP44-less centrioles did successfully accomplish the removal of the cartwheel (Fig. 5d).

The length of CEP295-less centrioles is reduced[15]. To exclude that length differences between CEP295- and CEP44-less centrioles explain the stability difference, we tested whether also CEP44-less centrioles had a defect in elongation. The centriole length was deduced from the diameter of rosette structures[34] of late G2 phase cells. Depletion of CEP295 or CEP44 generated similarly short dCs (Fig. 5e, f) excluding that length differences of centrioles are the explanation of the difference in centriole stability upon depletion of CEP295 and CEP44.

These data suggest that CEP295 functions as dC stabilising factor independent of its PCM recruitment and centriole elongation role, probably explaining the higher abundance of CEP295 on dCs than on the mCs[17] (Fig. 4f, Supplementary Fig. 10d).

**Conversion proteins provide structural integrity of new dCs.** Defects in centrosome biogenesis may be due to centriole maturation issues. Centriole MTs glutamylation is an index of centriole maturation and stability[35–37]. We therefore tested whether tubulin of non-converted centrioles underwent gluta-mylation. The G1 non-converted daughter centrosomes of CEP295, CEP44 or POC1B depleted cells were less stained by the glutamylation marker GT335 than control centrioles (Fig. 6a, b, Supplementary Fig. 11a, b for Cenp-F). This may indicate that reduced tubulin glutamylation triggers a CCC defect. However, overexpression of *CCP5*[38] that de-glutamylated most MTs within G1 centrioles (Supplementary Fig. 11c, d) had neither an impact on the PCM recruitment process (Supplementary Fig. 11c, e) nor on stability of centrioles in the cold (Supple-mentary Fig. 11f, g). Together, depletion of conversion factors

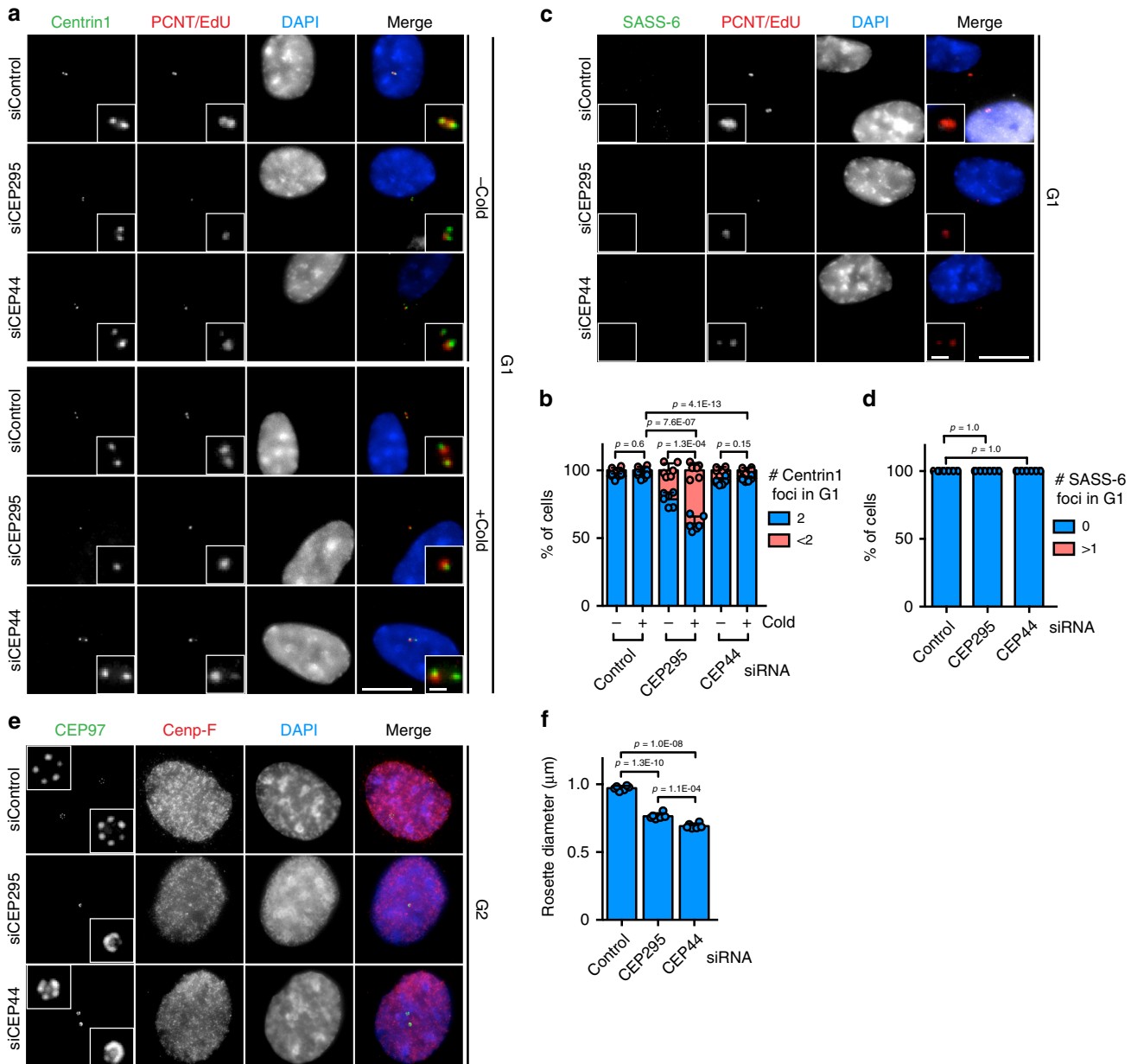

**Fig. 5 CEP295 stabilises new dCs. a** IF of siControl, siCEP295 and siCEP44 depletion samples untreated (upper 3 panels) and treated with cold (upper 3 panels). Similarly to control cells, in both cold treated and untreated siCEP44 samples G1 cells contained 2 centrioles (judged by centrin1 signals). In untreated siCEP295 cells, centrioles showed already a degree of instability (<2 centrin1 foci in 21.4 ± 5.0% of the G1 cells (Cenp-F in Supplementary Fig. 10b), which became more pronounced if the cells were exposed to cold treatment (<2 centrin1 foci in 39.3 ± 5.2% of the G1 cells (Cenp-F in Supplementary Fig. 10b). **b** Quantification of **a**. **c**, **d** SASS-6 is removed successfully in all (**d**, 100% of cells with no SASS-6 foci) G1 cells (Cenp-F in Supplementary Fig. 10c) in control, CEP44 and CEP295 siRNA samples. Note, the lower cell in the siControl is Cenp-F positive (Supplementary Fig. 10c) and so in G2 and therefore carries a SASS-6 signal. The enlargements in the siControl are from the upper Cenp-F negative cell. **e** IF of late G2 (Cenp-F positive and separated centrosomes) U2OS cells in which the expression of *Myc-PLK4* was induced upon siRNA treatment. In siCEP295 and siCEP44 cells the rosette generated upon *Myc-PLK4* overexpression showed a shorter diameter in comparison to control cells rosette. **f** Quantifications of **e**. Control rosette diameter 1.0 ± 0.1 μm, CEP295 depletion 0.7 ± 0.1 μm and CEP44 depletion 0.8 ± 0.1 μm. (**a**, **c**, **e**, scale bars: 10 μm, magnification scale bars: 1 μm; **b**, **d** and **f** data are presented as mean ± s.d., all statistics were derived from two-tail unpaired *t*-test analysis of *n* = 6 biologically independent experiments and source data are provided as a Source Data file).

affects the glutamylation of centriole MTs but this lack is not the cause of the CCC defect.

Because the glutamylation takes place specifically on the outer tubules of the centriole triplets[39], we hypothesised that that reduced tubulin glutamylation reflected centriole structural defects. To test this, we depleted CEP295, CEP44 or POC1B, arrested the cells in G1 by inhibiting Cdk4/6 and then analysed G1 centrioles cross-

sections by electron microscopy (Fig. 6c). We focused our analysis on the proximal end of the dC, the region where the triplets are present and where the CCC process happens. Cross-sections of the proximal end of control G1 dCs showed an intact structure (Fig. 6d). Differently, in the CEP295, CEP44 and POC1B depletion samples, cross-sections of the dCs showed that the triplet formation partially did not occur or the A-C linker was missing (Fig. 6e–g).

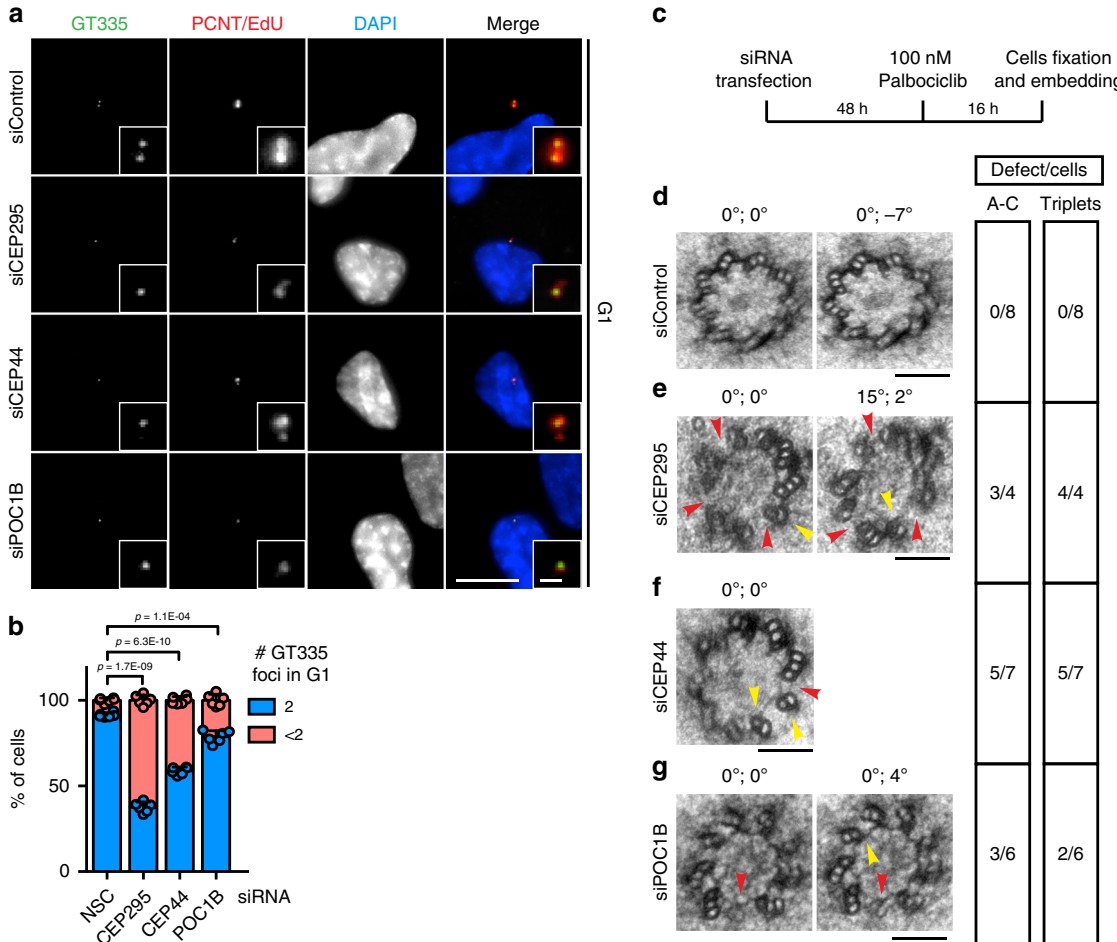

**Fig. 6 Conversion molecules provide structural integrity of centrioles. a** IF samples of cells treated with siControl, siCEP295, siCEP44 or siPOC1B. Depletions of siCEP295, siCEP44 or siPOC1B generated G1 cells (Cenp-F in Supplementary Fig. 11b) with <2 centriole glutamylation foci (GT335 signals) different from siControl. **b** Quantifications of **a**. 62.4 ± 3.0% of G1 siCEP295 cells showed <2 GT335 foci; 41.2 ± 2.2% in siCEP44 cells; 21.3 ± 3.5% in siPOC1B cells. Data are presented as mean ± s.d., statistics were derived from two-tail unpaired *t*-test analysis of *n* = 6 biologically independent experiments and source data are provided as a Source Data file. **c** Scheme of the experimental procedure of the EM analysis of dCs. **d–g** EM images of turned and tilted sections of proximal region of daughter centrosomes in G1. The two values on the top of the image give the sample turning and tilting degrees, respectively. Red arrowheads indicate open structural defects (A-C linker absent), while yellow ones highlight defects of triplets. The side table (right) shows quantification of the number of cells showing these defects. While in **d** G1 siControl cells there was no structural defect, in **e** siCEP295, **f** siCEP44 and **g** siPOC1B G1 cells dCs showed defects in the triplets and A-C linker formation. (**a** scale bar: 10 µm, magnification scale bar: 1 µm; **d–g**, scale bars: 100 nm).

We conclude, the depletion of the conversion molecules CEP295, CEP44 or POC1B commonly affect the architecture of the centriolar wall of centriole MTs.

**Structural integrity of the centrioles ensures their CCC.** TUBE1 (ε-tubulin isoform) and TUBD1 (δ-tubulin) were described to be important for MT triplet formation of centrioles[14]. As a proof of the concept that the formation of bona fide proximal end of the centriole structure is a requirement for the CCC, we depleted TUBE1 and TUBD1 via siRNA and analysed cross-sections of the dCs proximal end by EM. Differently from the control cells (Fig. 7a), the depletion of either TUBE1 or TUBD1 affected triplet formation (and TUBE1 depletion also in the A–C linker formation) (Fig. 7b, c), as previously reported in studies conducted in *Chlamydomonas reinhardtii*[40–44], *Paramecium terauralia*[45], and *Tetrahymena thermophila*[41]. The nature of the centriole defects was similar to the defects observed in centrioles of CEP295-, CEP44- and POC1B-depelted cells (Fig. 6e–g). However, the

siRNA depletion phenotype was milder than the reported centriole defect in ε- and δ-tubulin knockout human cell lines that assemble centrioles by the de novo pathway[14]. Residual amounts of ε- and δ-tubulin probably buffer the strength of the defect under siRNA conditions. Because of the lack of ε- and δ-tubulin antibodies and the failure of NeonGreen fused ε- and δ-tubulin to given a detectable signal, we were unable to confirm this notion.

We tested the presence of the centriole maturation marker GT335 in G1 centrioles of TUBE1 or TUBD1 depleted cells. In all, 25–35% of these centrioles did not undergo the glutamylation process (Fig. 7d, e, Supplementary Fig. 12a for Cenp-F). We next tested whether ε- and δ-tubulin–less centrioles had problems in recruiting PCM components to confirm that structural centriole defects impair the CCC. In both TUBE1 and TUBD1 siRNA-depleted cells, the G1 centrioles inefficiently recruited PCM (judged by counting number of defined γ-tubulin foci) (Fig. 7f, g, Supplementary Fig. 12b for Cenp-F). Analysis of dCs for glutamylation as a readout for centriole MT defects and γ-tubulin staining indicated that both defects occurred at the same

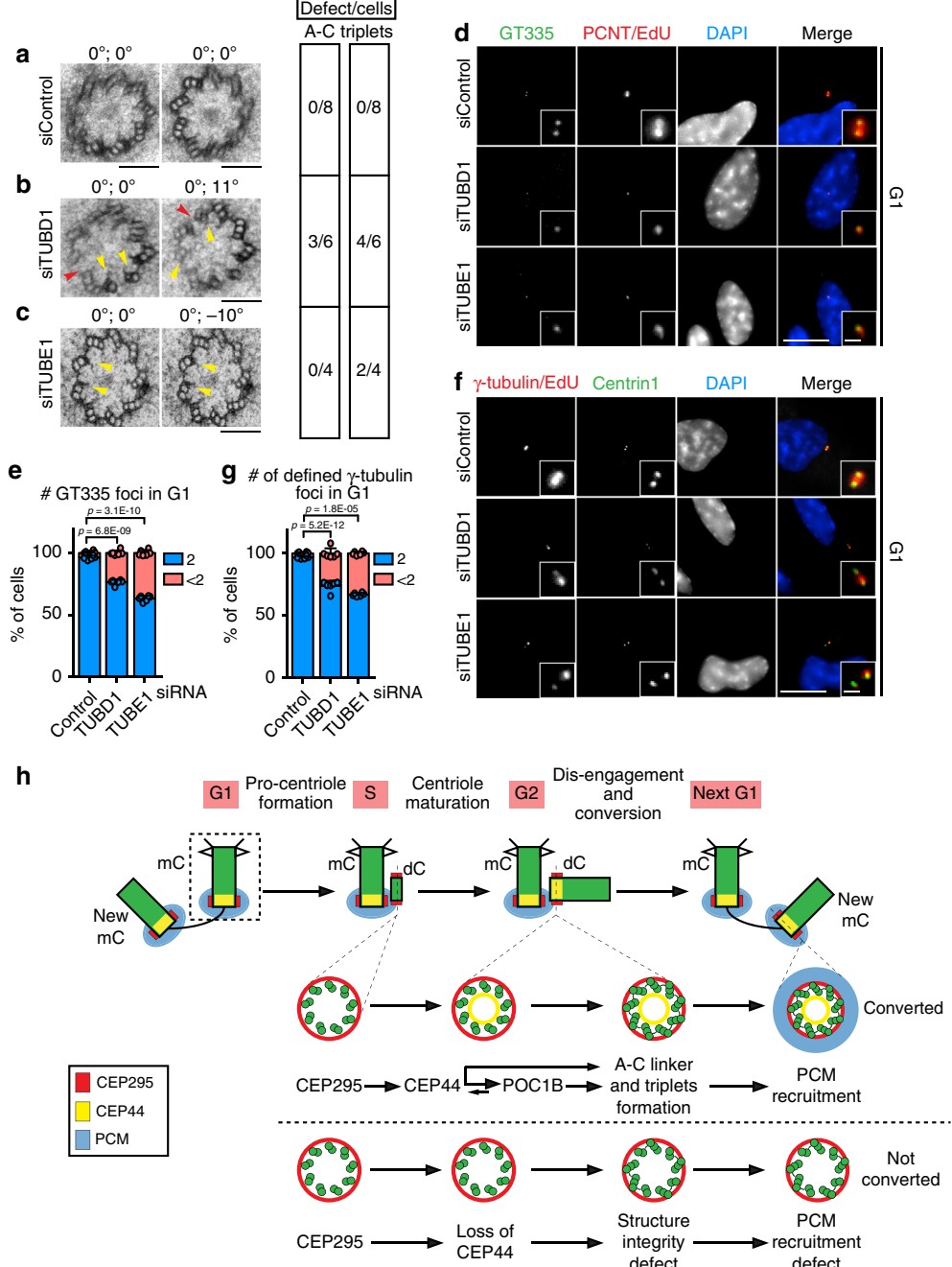

**Fig. 7 Structural integrity of the centrioles ensures their conversion to centrosomes. a–c** EM images of turned and tilted sections of proximal region of daughter centrosomes in G1. The two values on the top of the image give the turning and tilting degree of the sample, respectively. Red arrowheads indicate open structural defects (A–C linker absent), while yellow ones highlight defects of triplets. The side table (right) shows quantification of the number of cells showing these defects. While in **a** G1 siControl cells had no structural defects, in **b** siTUBD1 and **c** siTUBE1 G1 dCs showed defects in triplet formation (yellow arrow). **b** siTUBD1 G1 cell with defect in the A–C linker formation on top of the triplet formation of dCs and lack of triplet MTs (red arrow). **d** Depletion of TUBD1 and TUBE1 proteins also generated reduced glutamylation of the centrioles as judged by GT335 staining (36.7 ± 2.2% of G1 cells, Cenp-F in Supplementary Fig. 12a) with <2 GT335 foci in siTUBE1 sample; 23.5 ± 2.0% for siTUBD1). **e** Quantification of **d**. **f** In absence TUBE1 or TUBD1 many centrioles in G1 cells (Cenp-F in Supplementary Fig. 12b) did not efficiently acquire PCM (<2 γ-tubulin defined foci) to convert to centrosomes. **g** Quantification of **f**. 33.1 ± 1.3% of cells with <2 γ-tubulin foci in G1 for siTUBE1; 26.2 ± 4.1% for siTUBD1. (**a–c**, scale bars: 100 nm; **d**, **f** scale bars: 10 μm, magnification scale bars: 1 μm; **e**, **g** data are presented as mean ± s.d., all statistics were derived from two-tail unpaired t-test analysis of n = 6 biologically independent experiments and source data are provided as a Source Data file). **h** Model of the dependency of CCC mechanism on the correct centriolar wall maturation. See Discussion for description.

centrioles (Supplementary Fig. 12c). Interestingly, the centriole wall defects generated by the loss of ε- and δ-tubulin did not affect the early CCC component CEP295 (Supplementary Fig. 12d, e) but it did delocalise CEP44 and its binding partner POC1B (Supplementary Fig. 12f–i). ε- and δ-tubulin therefore function in between CEP295 and CEP44.

We conclude that a correctly assembled centriole is likely to be a prerequisite for the ability to recruit PCM to the new dC.

## Discussion

The current understanding of the molecular basis for CCC is based on data derived from studies in *D. melanogaster* and human cultured cell line. It was suggested that a protein bridge consisting of CEP135, CEP295 and CEP152/CEP192 extends from the centriolar wall to the outer PCM and promotes the recruitment of outer PCM proteins[15–17,46]. Therefore, surprisingly, loss of the centriole luminal protein CEP44 impaired CCC. The finding that CEP295 was still associated with daughter centrosomes of CEP44-depleted cells but was insufficient to promote CCC suggests additional mechanistic principles of CCC than the CEP295-dependent PCM recruitment cascade.

Here we describe a recruitment programme of centriolar proteins that starts with the CEP295, followed by CEP44 and POC1B (Fig. 7h). The connection between CEP295 and CEP44 is unclear since both proteins do not physically interact. Structural changes of centriolar MTs induced upon CEP295 binding to the procentriole probably allow the recruitment of CEP44. Consistent with this notion, depletion of TUBE1 and TUBD1 that affect C-MT biogenesis[14] also reduced CEP44 recruitment to centrioles.

CEP44 directly interacts with POC1B both in vivo and in vitro. In addition, our experiments suggest that POC1B targeting to the newborn centriole depends on CEP44 MT-binging ability. However, further studies indicated that both proteins showed a more complex localisation interdependency. While the co-localisation of both proteins within dCs is consistent with the CEP44-POC1B interaction, mature daughter centrioles only showed a partial co-localisation of CEP44 and POC1B. In addition, overexpression of NT-CEP44 that bound MTs and associates with centrioles but did not interact with POC1B was able to target a small fraction of POC1B to centrioles and to promote CCC to some level. We propose that NT-CEP44 via its ability to bind to A-MTs of centrioles impacts centriole structure and this then allows POC1B recruitment and CCC. However, our experiments also show that complete POC1B centriole recruitment and CCC is depended on the POC1B-binding region of CEP44. Based on these findings, we propose that transient CEP44-POC1B complex formation is needed early in centriole biogenesis to create a centriole structure that then allows CEP44-independent recruitment of POC1B.

Surprisingly, the localisation of CEP135 with dCs was also dependent on CEP44 despite the presence of CEP295 on dCs. This finding is different from what has been reported in *Drosophila*[16]. In human cells, CEP295-CEP135 are interdependently recruited to dCs[47]. We showed that CEP135 is not necessary for CCC, but its recruitment might require structurally-intact centrioles that are ensured by CEP295 and CEP44.

The arching function of CEP295, CEP44 and POC1B is the biogenesis of centrioles with a proper 9 MT triplet structure. Depletion of any of these proteins affected the structure of centrioles (Fig. 7h). Single C-tubules were either missing or the A-C linker was defective. To proof that mild centriole structure defects affect the CCC, we depleted via siRNA δ- and ε-tubulin isoforms. Similarly to CEP295, CEP44 and POC1B, depletion of δ- and ε-tubulin generated centrioles with moderate structural defects that impaired CCC. We propose that centrioles with even minor structural alterations will not or only partially undergo the CCC programme (Fig. 7h). This notion is consistent with the observation that even tiny changes in the 9-fold symmetric structure of the centriole can have a big impact. Small changes in the *Tetrahymena* basal body affected its ability to withstand the forces produced by motile cilia beating[48]. These relatively subtle defects probably affect the recruitment of proteins responsible for PCM organisation (e.g. CEP57[49]) that depend on the correct spacing of triplet MTs in centrioles. Alternatively, cooperative interaction and binding of inner PCM proteins, as recently indicated for the

assembly of CEP63 and CEP152 rings[50], may be affected when the centriole scaffold is remotely defective. Due to these key-lock properties of centriole-PCM recruitment, centrioles provide a unique platform in cells that is only recognised at physiological concentration by centriolar MT-binding proteins that do not bind to other MT structures.

Interestingly, centrioles lacking CEP295 were less stable than CEP44-less centrioles, to reveal an additional role of CEP295 as centriole stabilisation factor. This stabilising function of CEP295 becomes important upon removal of the SASS-6 cartwheel during mitosis[15], an event that does not happen in *D. melanogaster*[10]. The absence of a CEP44 homologue and the different interdependency between CEP135 and CEP295 in *D. melanogaster* versus in human cells suggest that CCC may differ between organisms. Interestingly, in the *D. melanogaster* germline stem cells, where POC1B plays an important role in centriole biogenesis, the centriole are composed of MT-triplets, differently from somatic cells where the MT-wall is composed of doublets[51–54].

In summary, we propose that after SASS-6 cartwheel assembly and formation of centriole MTs, a defined recruitment programme starting with CEP295 promotes centriole biogenesis (Fig. 7h). CEP295 binding is followed by the establishment of an integer MT wall structure through the loading of CEP44 and POC1B to the lumen of centrioles and the action of δ- and ε-tubulin probably in formation of the C-MTs[14]. This finding suggests that the developing centriolar structure functions as one of the most important requisites for recruitment of PCM proteins. It explains the devastating consequences of failed centriole biogenesis, since even minute defects of the centriolar structure can affect the CCC and therefore centrosomal function such as duplication and MT nucleation in the reaching context within which this organelle operates.

## Methods

**Cell culture**. Human telomerase-immortalised retinal pigment epithelial cells (hTERT-RPE1, RPE1) *CEP44*−/− clones and stable cell lines described in this work were cultured in DME/F-12 (1:1) medium supplemented with 10% foetal bovine serum (FBS), 1% L-Glutamine and 1% penicillin–streptomycin. human embryonic kidney 293 (T) (HEK T293) cells and HEK GP2-293 (Clonetech) cells were cultured in the same medium as RPE1 cells. U2OS tetracyclin-inducible cell-line expressing Myc-tagged PLK4[34] were cultured in DMEM medium supplemented with 10% FBS, 1% L-Glutamine, 1% penicillin-streptomycin, 1 mg/ml G418 (Invitrogen) and 50 μg/ml Hygromycin B.

**Drug treatments**. The Cdk4/6 inhibitor Palbociclib (Tocris Cat. #4786) was used to arrest RPE1 cells in G1 phase. Cells were treated 16 h with 100 nM of Palbociclib (Dr. E. Trotter, Prof. Dr. I. Hagan).

EdU from Click-iT™ Plus EdU Alexa Fluor™ Imaging Kit (ThermoFisher Cat. # C10638 and #C10640) was used to detect cells in S phase following manufacturer's protocol. To detect specifically S phase cells, 10 nM EdU was added to cells in culture 15–20 min prior to cell fixation.

**Plasmid transfection and RNAi**. Plasmid transfection into HEK GP2-293 was accomplished via PEI reagent. Transfection of synthetic siRNA oligos (Supplementary Table 1) into RPE1 cells as well as into U2OS cells was performed using Lipofectamine® RNAiMAX Transfection Reagent from Life Technologies. Transfection reactions were prepared in Opti-MEM™ medium following manufacturer's protocol.

**Plasmids and constructs**. pRetroX-Tet3G and pVSVG plasmids (Retro-X™ Tet-On® 3 G Inducible Expression System - Clontech) were used to generate RPE1 cells with Tet-On® 3 G System. siRNA-resistant *CEP44* cDNA was Flag-tagged via PCR and cloned into pRetroX-TRE3G. Because 2 of the 4 ON-TARGETplus siRNA SMARTpool oligos target the coding region of CEP44 mRNA, the cDNA was mutated in those two loci of the sequence. To make the transcript resistant to the siRNA, several mutations of the cDNA sequence were designed leaving the encoded amino acids unchanged: the *CEP44* cDNA sequence TTATCCTGAAGA GGT (position 78-93 nt) was mutated into cGAtTGcGTcGGacTt and the sequence CCAATGTGGGTTTGCAGAA (291-309 nt) into aCAgTGcGGcTTcGCcGAg (Supplementary Tables 2 and 3).

pX458 plasmid was used to generate *CEP44*<sup>−/−</sup> cell lines. The sgRNA targeting the exon 1 of *CEP44* gene was designed using the CRISPR Design tool of the Zhang Lab, MIT[27]. The oligos (Supplementary Table 3) were used to clone the sgRNA sequence into the pX458 plasmid (Zhang Lab, MIT).

Other cDNA integrated in RPE1 cells: *mCCP5-HA, TUBE1-Flag* and *TUBD1-Flag* tagged (DKFZ) (Supplementary Table 3).

**Stable cell lines and *CEP44*<sup>−/−</sup> cell lines.** For rescuing the CEP44 conversion phenotype, a cell line expressing a siRNA-resistant construct was made from RPE1 wild type cells. Firstly, the Retro-X™ Tet-On® 3G Inducible Expression System (Clontech) was introduced in RPE1 wt cells. siRNA-resistant constructs of *CEP44* were the integrated in RPE1 Tet3G cells under the TRE3G promoter via Retrovirus infection (Clontech) (Supplementary Table 3).

*CEP44*<sup>−/−</sup> cell lines were generated via electroporation of RPE1 *p53*<sup>−/−</sup> (Prof. Dr. Bryan Tsou) with pX458 cloned with the sequence for the sgRNA (Supplementary Table 3). Two days after electroporation GFP positive cells were FACS-sorted and the pool of cells obtained from the sorting was tested with antibodies against CEP44. In the knockout pool the signal of CEP44 vanished in 50% of the cells. Single clones were then tested by indirect immunofluorescence (IF), immunoblotting and chromosomal DNA sequencing to select cells in which both *CEP44* alleles were successfully knocked out. Two *CEP44*<sup>−/−</sup> genetically different clones were used for experiments. Clone #2 had a heterozygote *CEP44* genotype (2 nt deletion on one copy and 5 nt deletion on the other copy), generating a non-sense frame shift. Clone #7 is homozygote with 2 nt deletion on both gene copies (Supplementary Fig. 3).

**Immunofluorescence.** Cell were fixed on coverslips with methanol at −20 °C for 5 min and the coverslips blocked in 10% FBS, 0.1% Triton-X100 for 30 min. After incubation for 1 h with primary antibody (diluted in 3% bovine serum albumin, BSA (w/v)), cells were incubated with secondary antibody (1:500 dilution in 3% BSA) and 4′,6-Diamidine-2′-phenylindole dihydrochloride (DAPI) and mounted on glass slides with Moviol or ProLong™ Gold antifade mountant for super resolution microscopy. Cells were extracted with CSK-extraction buffer prior staining with indicated antibodies. The EdU stain detection reaction was accomplished after coverslip blocking and before primary antibody incubation, following the reaction instruction of Click-iT™ Plus EdU Alexa Fluor™ Imaging Kit (555 and 647 Kits, ThermoFisher).

Images were acquired on a DeltaVision RT system (Applied Precision) with an Olympus IX71 microscope equipped with 60X and 100X objective lenses. SIM images were acquired on N-SIM microscope equipped with ×100 objective lenses (Nikon).

**Antibodies.** CEP44 antibodies were raised in rabbits immunised against the CEP44 C-terminal-6xHis-tagged protein purified from BL21-CodonPlus™ bacteria transformed with pET-28c (+)-*CEP44* plasmid vector (Supplementary Table 3). The antibody was affinity purified from the serum with antigen-coupled CNBr sepharose.

POC1A and POC1B antibodies were raised in guinea pigs, which were immunised against either POC1A (aa 296-369) or POC1B (aa 304-422) peptides tagged with a C-terminal-GST purified from BL21-CodonPlus™ bacteria transformed with pGEX-6P-1-*POC1A/B* plasmid vectors (Supplementary Table 3). The antibodies were affinity purified from the serum with antigen-coupled CNBr sepharose.

Other primary antibodies directed against the indicated proteins were: γ-tubulin (mouse, 1:1000, abcam Ab27074), PCNT (rabbit, 1:2000, abcam Ab4448), CEP97 (rabbit, 1:300, Bethyl A301-945A), Centrin1 (mouse 1:1000, Millipore MABC544), α-tubulin (mouse, 1:500, SigmaAldrich DM1A), SASS-6 (mouse, 1:50, SCBT sc-81431), Flag tag (mouse, IF 1:1600 - WB 1:1000, Cell signaling 9A3), CEP295 (rabbit, 1:500, Abcam Ab122490), GAPDH (rabbit, WB 1:1000, 14C10), HA tag (rat, 1:1000, Böhringer Mannheim No.1867423), GT335 (mouse, 1:500, AdipoGen AG-2013-0020), γ-tubulin (guinea pig, 1:50), CEP135 (rabbit, 1:100), C-Nap1 (goat, 1:1000), CEP164 (rat, 1:2000), CEP152 (rabbit, 1:500, Dr. Ingrid Hoffmann, DKFZ), CEP192 (rabbit, 1:2000, Nigg) and Cenp-F (sheep, 1:1000, Dr. Stephen Taylor, Manchester).

**Protein purification.** *CEP44-Flag, CEP44-Flag h5*<sup>-</sup> and *POC1B-HA* recombinant constructs were cloned into pGEX-6P-1 vector containing an in frame-cleavable C-terminal GST and transformed into *E.coli* BL21-CodonPlus™ (Supplementary Table 3). The protein expression was induced with 0.4 mM Isopropyl β-D-1-thio-galactopyranoside (IPTG) and the bacteria were harvested after an overnight incubation at 20 °C. Upon sonication-induced cell lysis in 250 mM NaCl, 50 mM Tris-Cl, 1 mM EDTA, 1.2 mM PMSF, 1 mM DTT, 1 Tablet cOmplete™ Protease Inhibitor Cocktail (SigmaAldrich 11697498001) buffer, the protein was bound to Protino® Glutathione Agarose 4B beads (Macherey-Nagel 745500.100) and eluted with 50 mM reduced glutathione (GSH). The eluted proteins were dialysed overnight and the GST was cleaved off by GST-TEV protease. To remove GST and TEV-GST, a second incubation with Protino® Glutathione Agarose 4B beads was performed. The flow through was collected and the purity of the proteins was analysed by SDS-PAGE and Coomassie Blue staining.

**Microtubule binding assay.** MTs were polymerised from purified pig-brain tubulin for 30 min at 37 °C in BRB80 buffer (80 mM PIPES-KOH pH 6.8, 1 mM MgCl₂, 1 mM EGTA) in presence of 1 mM GTP, before being stabilised through the addition of 10 μM Taxol at room temperature (RT) for 15 min and pelleted (195.000 × *g* for 20 min at 37 °C). The pellet was re-suspended in a solution containing either GST-Flag, CEP44-Flag or h5<sup>-</sup> CEP44-Flag recombinant proteins at a final concentration of 1 μM and the mixture was incubated for 20 min at RT prior centrifugation (195.000 × *g* for 20 min at 37 °C) to separate pellet and soluble fractions that were analysed by immunoblotting (IB).

**CEP44-Flag IP.** In order to test CEP44 protein interactions in RPE1 cells, *CEP44-Flag* was expressed under control of the tetracycline inducible promoter in RPE1 cells and the lysate of 3 × 10⁶ cells was incubated together with Anti-FLAG® M2 Beads (SigmaAldrich A2220; Fig. 3a, e and Supplementary Table 1). The lysis of RPE1 cells was accomplished by pipetting the harvested cells in 250 mM NaCl, 10 mM Tris-Cl, 0.5 mM EDTA, 0.5% NP-40, 1 mM PMSF, 10 U/μl Benzonase, pH 7.5 buffer. The lysate was then incubated with the beads for 2.5 h with shaking at 4 °C. After three washing steps with 150 mM NaCl, 10 mM Tris-Cl, pH 7.5 buffer, the bound proteins were eluted in 30 μl of 4x Laemmli buffer 5 min 95 °C. The samples were analysed by IB.

The sample preparation for the IP-mass spectrometry experiment (Supplementary Table 1) followed the CEP44-Flag IP protocol above but differed in the amount of the starting material and lysis condition (30 × 10⁶ cells were lysed in 150 mM NaCl, 10 mM Tris-Cl, 0.5 mM EDTA, 0.5% NP-40, 1 mM PMSF, 10 U/μl Benzonase, pH 7.5 buffer). Proteins were eluted from the Anti-FLAG® M2 Beads by three steps of incubation with Flag peptide (3X FLAG® Peptide, SigmaAldrich F4799-4MG). Eluted proteins were concentrated by trichloroacetic acid (TCA) precipitation and then analysed by mass spectrometry.

**CEP44-POC1B in vitro binding assay.** CEP44-Flag and POC1B-HA recombinant proteins were purified from *E.coli* separately (Supplementary Fig. 5c) (Supplementary Table 3). To test in vitro binding between CEP44 and POC1B, 2 μM POC1B-HA was incubated with either empty Anti-FLAG® M2 Beads (SigmaAldrich A2220) or in presence of 2 μM recombinant CEP44 in 150 mM NaCl, 50 mM Tris-Cl pH 7.5 buffer. After 2 h incubation (4 °C), the beads were washed 3x with the 150 mM NaCl, 50 mM Tris-Cl pH 7.5 buffer and the protein was eluted in 30 μl of 4x Laemmli buffer 5 min 95 °C. The samples were analysed after SDS-PAGE by Coomassie Blue staining and IB.

**Microtubule regrowth assay.** For MT regrowth assay, cells were first transferred on ice for 30 min before being washed with 37 °C 1X PBS and transferred quickly to a 37 °C pre-heated metal block for 5 s. To enhance MT visualisation, the cells were briefly extracted with CSK buffer and fixed with methanol.

**Centriole stability assay.** Cells were transferred to sit on ice for 1 h prior fixation.

**Mass spectrometry.** The sample preparation for the IP-mass spectrometry experiment is described in the method section CEP44-Flag IP. The mass spectrometry and its data analysis were done comparing the IP sample of cells expressing CEP44-Flag recombinant protein upon doxycycline induction and the IP sample of RPE1 Teton wild type cells also induced with doxycycline, used as control. The mass spectrometry experiment represents *n* = 1 biological and technical replicate. The interaction hits of our interest were validated with further independent biological experiments.

The IP samples were run 1 cm on a 10% SDS-PAGE gel. The gel pieces in which the samples were present were excised and reduced with 60 μl 40 mM dithiothreitol (DTT; Sigma-Aldrich, Taufkirchen, Germany) in 50 mM TEAB, pH 8.5 at 57 °C for 30 min and alkylated with 60 μl 50 mM iodoacetamide (IAA; Sigma-Aldrich, Taufkirchen, Germany) in 50 mM TEAB, pH 8.5 at 25 °C for 20 min in dark. Gel pieces were dehydrated with 60 μl 100% ACN and washed with 60 μl 50 mM TEAB, pH 8.5. A total of 30 μl of 8 ng/μl in 50 mM TEAB trypsin solution (sequencing grade, Thermo-Fisher, Rockford, USA) was added to the dry gel pieces and incubated for 6 h at 37 °C. The reaction was quenched by addition of 20 μl of 0.1% trifluoroacetic acid (TFA; Biosolve, Valkenswaard, The Netherlands). The resulting peptides were extracted by dehydrating two times for 20 min each in 20 μl ACN and washing in 30 μl 50 mM TEAB, pH 8.5. The supernatant from each extraction step was collected and dried in a vacuum concentrator before dimethyl labelling reaction.

Dimethyl duplex labelling was performed according to standard protocol[55]. Briefly, digested peptides from samples were tagged in solution with stable-isotope dimethyl labels comprising regular formaldehyde and cyanoborohydride (28 Da shift, designated "light label") or deuterated formaldehyde and regular cyanoborohydride (32 Da shift, designated "intermediate label") (all reagents from Sigma-Aldrich, Taufkirchen, Germany). Samples were acidified with TFA such that the pH < 2 and desalted with C18 stagetips[56]. Samples were dried in a vacuum centrifuge and stored in −20 °C until measurements. Samples were diluted in 15 μl 0.1% TFA, 99.9% water for LC-MS measurement.

Nanoflow LC-MS² analysis was performed with an Ultimate 3000 liquid chromatography system directly coupled to an Orbitrap QE HF spectrometer (both

Thermo-Fischer, Bremen, Germany). Samples were delivered to an in-house packed analytical column (inner diameter 75 μm × 25 cm; CS – Chromatographie Service GmbH, Langerwehe, Germany) filled with 1.9 μm ReprosilPur-AQ 120 C18 material (Dr. Maisch, Ammerbuch-Entringen, Germany). Solvent A was 0.1% formic acid (FA; ProteoChem, Denver, CO, USA) and 1% ACN (Biosolve) in $H_2O$ (Biosolve). Solvent B was composed of 0.1% FA (ProteoChem), 10% $H_2O$ (Biosolve) and 89.9% ACN (Biosolve). Sample was loaded to the analytical column for 20 min with 3% B at 550 nl/min flow rate. The peptide separation was carried out with 120 min linear gradient (3–38% B) with the reduced flow rate of 300 nl/min. The mass spectrometer was operated in data-dependent acquisition mode, automatically switching between MS and MS[2]. MS spectra ($m/z$ 400–1600) were acquired in the Orbitrap at 60,000 ($m/z$ 200) resolution. Fragmentation in HCD cell was performed for up to 15 precursors with normalised collision energy of 27%. MS[2] spectra were acquired at 30,000 resolution ($m/z$ 200).

Raw files were processed using MaxQuant v1.5.3.30[57] for peptide identification and quantification. MS[2] spectra were searched against the Human_201706_UP000005640 _9606 (20147 sequences) and Human_201706_UP000005640_9606_additional (122129 sequences) retrieved in June 2017 and contaminants of Maxquant. We defined carbamidomethylation of cysteine residues as fixed modification and acetyl (protein N-term), oxidation of methionine, deamidation of aspartic and glutamic acid as variable modifications and trypsin/P as the proteolytic enzyme with up to two missed cleavages allowed. The maximum false discovery rate for proteins and peptides was 0.01 and a minimum peptide length of seven amino acids was required. Match between run option was not activated. Quantification mode was with the dimethyl Lys 0 and N-term 0 as light labels and dimethyl Lys 4 and N-term 4 as intermediate labels. All other parameters were default parameters of MaxQuant. Quantitative normalised ratios were calculated by MaxQuant and used for further data analysis.

**Centrosome preparation and immuno-electron microscopy**. Centrosomes were purified from HEK T293 cells by following centrosome preparation protocol[58]. Fractions of the purification were tested by IB and IF prior immuno-gold labelling.

Isolated centrosomes were spun down onto coverslips and fixed on them with a solution of 4% paraformaldehyde, 0.1% glutaraldehyde, 2% sucrose in 100 mM phosphate buffer for 10 min at RT. After rinsing the aldehydes were quenched for 10 min with 50 mM glycine in PBS. The sample was blocked in 1.5% BSA for 20 min, incubated with primary antibody for 20 min before labelling with protein A gold (10 nm). After extensively rinsing, the centrosomes were briefly fixed again in 2% GA in 50 mM cacodylate solution for 15 min at RT. Rinsing in cacodylate buffer was followed by incubation for 40 min in 2% osmium/cacodylate buffer, on ice and in darkness. The samples were stained overnight at 4 °C with 0.5% uranyl acetate (in $H_2O$). On the following day coverslips were dehydrated by ethanol in a stepwise manner. Coverslips were immediately placed on capsules filled with Spurr-resin and polymerised for 24 to 48 h at 60 °C.

Embedded centrosomes were sectioned using a Reichert Ultracut S Microtome (Leica Instruments, Vienna, Austria) to a thickness of 70 nm. Post-staining with 3% uranyl acetate in $dH_2O$ and lead citrate was performed. Sections were imaged at a Jeol JE-1400 (Jeol Ltd., Tokyo, Japan), operating at 80 kV, equipped with a 4k × 4k digital camera (F416, TVIPS, Gauting, Germany).

**Electron microscopy analysis of dC proximal end**. Cells were seeded on carbon coated Sapphire discs (diameter 3 mm, thickness 0.05 mm, Engineering Office M. Wohlwend GmbH, Sennwald, Switzerland). After Palbociclib treatment, the cells were high-pressure frozen (HPF) using HPM 010 (Bal-Tec, Liechtenstein). For HPF the Sapphire with cells was sandwiched between two aluminium planchettes (Engineering Office M. Wohlwend GmbH, Sennwald, Switzerland) with the 0.1 mm deep aluminium carrier placed over the cells and the flat surface of the 0.3 carrier below the sapphire.

Freeze substitution of the cells was done by using a freeze substitution device EM-AFS2 (LeicaMicrosystems, Vienna, Austria). The freeze substitution solution contained 0.2% (w/v) osmium, 0.1% (w/v) uranyl acetate and 5% water (v/v) in dry acetone and the samples were substituted at −90 °C for 1 h. The temperature was then increased at a rate of 5 °C/h to 20 °C followed by 1 h incubation at 20 °C. Samples were rinsed with acetone followed by stepwise (30%, 50%, 75% and 100%) infiltration in spurr's resin (Serva, Heidelberg, Germany). Resin polymerisation was done at 60 °C for 48 h.

Sectioning was done on a Leica UC6 microtome (Leica Microsystems, Vienna, Austria) and 200-nm-thick serial sections were collected on Formvar-coated, copper slot grids. The grids were placed in a high-tilt holder (Model 2040; Fischione Instruments; Corporate Circle, PA) and images were taken on a Tecnai F20 EM (FEI, Eindhoven, The Netherlands) operating at 200 kV using the SerialEM software package[59] and an FEI Eagle 4K × 4K CCD camera.

All the centrioles were examined by rotating and tilting the grid to an angle where the centriole could be viewed through the long axis to reveal a cross section across the centriole barrel. The presence of appendages was checked by examining the serial sections of the two centriols in each pair. Based on this examination the dC lacking appendages was selected for imaging.

**Statistics and reproducibility**. Data presented in the Figs. 1a, 3a, b, e and 4a–d, f, h, and Supplementary Figs. 1l, 2b, 3h, 5c, d, 7a, 8a, b, e–g, 9b, c and 11c show

representative images of at least three biologically independent experiments giving reproducible similar results.

**Reporting summary**. Further information on research design is available in the Nature Research Reporting Summary linked to this article.

## Data availability

All data that support the findings of this study are available as Source Data File. The source data underlying all the quantification presented in this work are available either in Source Data file and Supplementary Information. The mass spectrometry proteomics data have been deposited to the ProteomeXchange Consortium via the PRIDE[60] partner repository with the dataset identifier PXD017332. GeneTree alignment is available under ENSGT00390000009873. All other relevant data supporting the key findings of the study are available from the corresponding author upon reasonable request.

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

## Acknowledgements

We thank Profs. Drs. E. Nigg (Basel, Switzerland), I. Hagan (Manchester, UK), B. Tsou (Rockefeller, USA), I. Hoffmann (DKFZ, Germany) and S. Taylor (Manchester, UK) for antibodies and cell lines used in this study. The electron microscopy was performed at the EM Core Facility at Heidelberg University and we are grateful for their support. We thank the Nikon Imaging Center of the University of Heidelberg and the Core Facility for Mass Spectrometry & Proteomics (CFMP) of the ZMBH at Heidelberg University for technical support. This work was supported by a grant of the German Research Council (DFG; Schi295/8-1).

## Author contributions

E.S.A. performed most of experiments. S.H. contributed in input for experiments and for manuscript. C.F. developed the EM analysis of daughter centrosome proximal end. A.N. performed immuno-EM and helped in the EM analysis of daughter centrosome proximal end. E.S.A. designed all the experiments with input of E.S. and S.H.; E.S.A. and E.S. wrote the manuscript.

## Competing interests

The authors declare no competing interests.
