## [Peer Review File · Nature Communications]

Reviewers' comments:

Reviewer #1 (Remarks to the Author):

In this manuscript, Atorino and colleagues identified a new regulator, CEP44, for the centriole-to-centrosome conversion (CCC). It has been previously reported that the occurrence of CCC is dependent on the function of CEP295. The authors found that CEP44 acts downstream of Cep295 in the process of the CCC, although the physical interaction between the two proteins is not clear. Depletion of CEP44 in human cells leads to defects in assembly of PCM and also, to some extent, centriole formation. The authors provided some evidence that CEP44 forms a complex with POC1B which is suggested to act as a part of the A-C linker between triplet MTs. Consistently, CEP44 colocalizes with POC1B at the proximity of the centriole lumen. According to the analogy of the N-terminal region of CEP44 with EB1 and 3, the authors generated a presumable MT-binding-deficient mutant of CEP44 and demonstrated that the mutant did not rescue the phenotype of CEP44 depletion. Interestingly, the stability assay with cold treatment revealed that CEP295 is more important to ensure the stability of newly-formed centrioles than CEP44. The authors further addressed the relationship between structural integrity of centrioles and the CCC, especially focusing on the arrangement of triplet MTs and the A-C linker. Using siRNAs against TUBD1 and E1, they show that the structural integrity of new centrioles is critical for the modification of triplet MTs and the CCC.

Overall, the quality of data is high and convincing, and the manuscript is well written. This manuscript would provide a new concept that the integrity of centriole wall is essential for proper CCC. Thus, this study will be of great interest to the centrosome field and also to cell biologists, and is therefore a strong candidate for publication in Nature Communications. However, there are several concerns that should be experimentally addressed in prior to publication.

Main points:

1. In Figure 1, to confirm the defect in the CCC upon CEP44 depletion, it would be better to distinguish between old and new mother centrioles using markers such as ODF2 and Cep164. Defects in the CCC should be more frequently detected on new mother centrioles.
2. In Figure2, if the authors intend to address the conversion mechanism, the centriole marker such as centrin, CP110 and CEP97 should be used in this figure. γ -tubulin is not an appropriate marker because its signal is also reduced in the absence of CEP44.
3. For example, in Figure2i, this reviewer could not understand why the number of CEP135 foci was significantly decreased upon CEP44 depletion. Fu et al (NCB, 2015) reported that CEP135 acts upstream of CEP295 in loading this protein to centrioles. Also, previous studies showed that CEP135 is critical for daughter centriole formation. As mention in the text, CEP135 is not a PCM protein. Considering the background of CEP135 function, this reviewer is confused with the result shown in Fig. 2i. The authors should clarify the cause of this phenotype; does the absence of CEP135 simply

reflect a decrease in the number of daughter centrioles? Otherwise, as the authors claimed, does this phenotype reflect defects in the CCC although daughter centrioles are somehow formed?

4. In Figure 3, the biochemistry testing the physical interaction between CEP44 and POC1B is rather weak. More fragments of CEP44 should be examined to narrow down the CEP44 domain responsible for specifically binding to POC1B, but not POC1A. In addition, to confirm the interaction, *in vitro* binding assay with the purified proteins could be useful. If possible, this result will be an important information for understanding the molecular architecture of basal part of centrioles.

5. In Figure 4, the phenotype that the CEP44 h5- mutant did not localize to centrioles is interesting. However, as this mutation seems to be predicted and designed based on its analogy with EB1 and 3 domains, the ability of h5- mutant protein for binding to tubulin should be directly tested by biochemistry as in Figure 4h. If the binding of CEP44 to tubulin is needed, how would Cep295 loss affect the loading of CEP44 to centrioles? because of defects in the centriole integrity? This possibility can be tested with siTUBD1 and siTUBE1. Also, did the authors test whether the N-terminal part of CEP44 is sufficient for its loading to centrioles?

6. In Figure 6, Venoux et al (2013, JCS) reported that POC1A and B act together to ensure the centriole integrity. Does double-knock-down of both proteins lead to more significant defects in the centriole structure and CCC? At least, IF-based experiments testing the loading of CEP295, CEP44, GT335, PCM proteins should be done in this condition.

7. In Figure 7, this is an excellent experiment addressing the effect of centriole structure defect solely on the microtubule modification and γ -tubulin loading. Using siTUBD1 and siTUBE1, the loading of CEP295, CEP44 and POC1A,B to centrioles should be tested. This experiment may address whether the pathway of CEP295-CEP44-POC1B actually works for the CCC mechanism, or the structural defects of centrioles by CEP295 depletion just affects the loading of CEP44 and POC1A,B and other microtubule binders.

Minor points:

8. In Figure 3, it would be more informative to indicate the alignment of CEP44 family proteins in vertebrates to see the evolutionarily conserved and functional domains, since this protein family is not well characterized thus far.

9. In Figure 5, this reviewer wonders how the cold treatment disrupts unstable or immature centrioles. Is this because of lack of tubulin modifications on centriolar microtubules upon CEP295 depletion? But, this might not be the case based on the result from Fig. 6a-b.

10. Would the CCC completely depend on the structural integrity of centriole wall? Otherwise, is there a mechanism separate from it? It would be interesting if the authors could discuss this issue with their ideas in the revised manuscript.

Reviewer #2 (Remarks to the Author):

Centriole-to-centrosome conversion (CCC), which renders daughter centrioles competent for motherhood, is required for the procentriole to acquire competence for duplication. While it is well appreciated that centriole maturation and CCC occur simultaneously, the relationship between the formation of centriole MT triplets and the recruitment of pericentriolar materials (PCM) remains largely unknown. In this manuscript, Atorino et al showed that assembly of normal centriole structure is critical for promoting timely CCC and generating functional centrosomes. Furthermore, they suggest that Cep44, a component of centriolar lumen, contributes to CCC by interacting with POC1B and aiding the recruitment of PCM components, such as Cep152, Cep192, PCNT, etc.

Overall, the authors have done a lot of work to understand how CCC is regulated and how the structure of centriole wall influences this process. Various knockdown/knockout analyses were carried out to delineate the CCC pathway. However, the drawback of this study is the lack of understanding at molecular levels. In addition, the analyses of knockdown cells are not rigorous {there are no data showing the levels of knockdowns by IB (except Cep44) or intensity measurements for controls; see below}. Likewise, whether the Cep44-POC1 axis mediates CCC in a bifurcated or parallel (i.e., independent) pathway remains elusive. Partially delocalized Cep44 by siCep295 and similarly delocalized Cep152, Cep192, and Cep135 by siCep44 would make it difficult to convincingly disentangle various components and their networks that contribute to the CCC pathway.

Major concerns:

1. Fig. 2, Supplementary Fig. 3c-h – As pointed out above, to delineate the Cep295-mediated CCC pathway, the authors should carry out more rigorous analyses with proper controls for side-by-side comparison. Notably, Cep295 localizes at the periphery of a centriole, whereas Cep44 localizes in the centriole lumen (as shown in Fig. 4). Thus, as the authors stated in line 347, how Cep295 can function at the upstream of luminal Cep44 remains a mystery. One possibility is that Cep295 mediates a bifurcated pathway and one of its branches is regulated by Cep44. Alternatively, Cep295 and Cep44 may mediate independent pathways that function in parallel to contribute to CCC. Unfortunately, no IB data are provided except the Cep44 IB shown in Supplementary Fig. 1a, thus making it difficult to properly interpret the data. Since depletion of one component may influence the stability of other components in the CCC pathway, performing IBs for each component in a way that allows cross-examination of all other components in the pathway would be very helpful. In addition, determining the severity of siCep295 siCep44 double knockdowns in comparison to

siCep295 or siCep44 alone will help propose whether the pathway is bifurcated or is composed of two independent pathways functioning in parallel to regulate the CCC. In the case of parallel pathways, Cep44 does not function at the downstream of Cep295.

Furthermore, the intensity measurements shown in Supplementary Fig. 3 lack important controls. For instance, for Supplementary Fig. 3d, the authors should provide the level of Cep295 intensity depleted by siCEP295 under the same conditions. This may allow one to assess whether ~50% reduction in Cep44 signal intensities achieved by siCEP295 is meaningful. If Cep295 depletion were as complete as for Cep44, then the ~50% reduced Cep44 signal could be resulted from gross structural defects associated with Cep295 depletion. Given the essential role of Cep295 in organizing a functional centrosome (Tsuchiya Y, et al, Nat Comm, 2016; Fu, J, et al, NCB, 2016), this point is especially important. Likewise, quantified Cep44 signal intensities for both siControl and siCep44 cells must be shown side-by-side to relatively assess the significance of ~50% reduced signal intensities for Cep152, Cep192, and Cep135 shown in Supplementary Fig. 3f-h. At present, it is premature to suggest the importance of the Cep44-POC1B axis in regulating downstream PCM proteins, such as Cep152, Cep193, and PCNT.

2. Fig. 3 – the authors show that Cep44 interacts with POC1B. However, with the data provided in Fig. 3b, apparently carried out using recombinant proteins as affinity ligands, it is difficult to judge how efficiently they interact with each other. If it is co-IP analysis, then input % needs to be shown. The lack of sufficient colocalization shown in Figs. 4d and 4f strongly suggests that these two proteins may not form a stable complex. Therefore, their partially interdependent colocalization shown in Fig. 4e-h could be due to a structural defect in centrioles by Cep44 RNAi. To properly assess the data, the authors should provide either IBs or quantified signal intensities for control and RNAi cells.

3. Fig. 4 – The authors nicely showed that the Cep44-h5 mutant's defect in MT binding cripples the CCC. In the light of one of their major findings that Cep44 interacts with POC1B to regulate the CCC (Fig. 3), generating a Cep44 mutant defective in POC1B binding could be more meaning for this work. Interestingly, unlike the MT plus-end binding EB1, Cep44 localizes to the centriole lumen under physiological conditions. Therefore, the authors may explore whether the capacity of Cep44 to interact with the lumen-localizing POC1B helps target Cep44 to this location. This notion can be discussed in the Discussion section.

4. fig. 6 – IB or quantified signal intensities showing the levels of knockdowns should be provided for better assessment of the data. If the levels of knockdowns are similar between Cep44 and POC1B, then these two proteins may not be in the linear pathway, as proposed in Figs. 3i and 7h. Rather, the delocalization of POC1B in Cep44 RNAi cells could be due to an indirect consequence of Cep44 RNAi-induced structural defects.

5. Supplementary Fig. 4g – Again, the normalized level of Cep44 signal intensities in siCep44 cells is necessary to comparatively assess the significance of the diminished POC1B signals in siCep44 cells.

If the Cep44 depletion is near complete, as shown in Supplementary Fig. 1a, then ~50% reduction in the POC1B signal may suggest that the Cep44-POC1B interaction in Fig. 3 is less likely significant.

6. Supplementary Fig. 6 – The data show that Cep295 preferentially localizes to daughter centrioles and functions as dC stabilizing factor. The author should examine whether this is the case for Cep44 and POC1, as suggested in their model.

Minor comments:

1. Fig. 4 – Schematic diagrams for the localized Cep44 and POC1B signals in 4d and Cep295 and tubulin signals at the daughter centriole in 4f will be helpful

2. Supplementary Fig. 1 – ~30-60% defect in the recruitment of PCNT and gamma-tubulin after a near-complete depletion of Cep44 suggests that Cep44 controls only a part of the CCC pathway.

3. Line 800 – the distance of “C- and B-tubule” should be changed to “B- and C-tubule”.

4. Line 91 – eliminate “)” from “---Cenp-F))”.

5. Line 135 – downstream “of” CEP295

Reviewer #3 (Remarks to the Author):

Review paper

General comments:

The authors attempt to demonstrate that successful centriole-to-centrosome conversion (CCC) relies on perfect structural integrity of the centriole. The authors focus their study almost exclusively on the study of the essential, but otherwise uncharacterised, protein CEP44 and its interaction with other CCC proteins. Why do the authors focus on this particular protein in the context of CCC

integrity? Since there is a multitude of proteins involved in the CCC process, their choice of CEP44 should be much more clearly justified.

The authors report a number of interesting individual findings, based on good experiments and mostly sound interpretation. However, the paper is poorly structured and the logic is not very well presented. We feel the paper should be presented differently (the title is far too general). It should be clear that this is a study on CEP44's potential role in CCC primarily, not on the centriole structural integrity's importance for CCC. It would be more appropriate to argue that centriole structural integrity offers a potential explanation for observed CEP44 phenotypes.

We would suggest the paper be structured along these lines (figure 5 does not fit in well with the rest of the paper and should not be presented here):

Introduction

- There are gaps in our understanding of CCC, which may be explained by the involvement of uncharacterised proteins.
- One such uncharacterised protein is CEP44, which has been suggested to be a centriolar protein in a published screen.
 - o Why CEP44, out of this list?

Results

- Figure 1: CEP44 is indeed a centriolar protein, and it is essential for CCC
- Next question: what is its role in CCC?
- Figure 2: CEP44 influences the recruitment of proteins downstream of CEP295, but not of CEP295 itself.
 - o Rephrase the figure title! Influencing a downstream pathway is not the same thing as being downstream in a pathway.
 - Next question: is it a component of this pathway (downstream)?
 - Supplementary table 1: CEP44 interactor analysis shows only POC1A and POC1B (?), not CEP295
 - o Were these really the only hits?
 - Figure 4: evidence of different localisation -> need to mention this here
 - o (Note this needs to be addressed more fully in the discussion, particularly in the last paragraph)
 - Next question: if CEP44 is not involved in the CEP295 pathway, does it interact with other characterised centriolar proteins/pathways? Considering its localisation to the centriole lumen, which proteins are attractive candidates and what does this suggest regarding potential functions (in structure)?
- Figure 3: CEP44 interacts with POC1B, and this complex is needed for CCC

- Next question: what is the role of the complex? Considering POC1B has a role in centriole maintenance, is it structural?
- o The leap between this complex and the investigation of the role of centriole structural stability in CCC needs to be made clear and explicit.
- Figure 6: conversion molecules are needed for structural integrity, including CEP44 and POC1B
- Next question: can this role account for the phenotype? Is it a potential explanation of the effect of CEP44 depletion on CCC?
- Figure 7: comprising centriole structural integrity via interference with tubulin epsilon and delta phenocopies CEP44 depletion

Conclusion

- The effect of CEP44 depletion observed in figure 1 may be due to it compromising centriole structural integrity.
- CEP295 cannot recruitment its downstream proteins if this structure is compromised.
- Therefore, CCC can be compromised (in disease) by loss of function of more proteins than just those involved in the key CCC pathway downstream of CEP295.

Specific comments:

The figures are not consistent in their lay-out. For instance, in figure 2, the colour scheme of the merged images is inconsistent, with the nucleus (DAPI stain) only being blue in some.

Much of the figures' content is presentation of single representative images. This is sometimes accompanied by quantification of a larger dataset, but this is missing for the intensity profiles presented in figure 4. How reproducible are these graphs?

In figure 4, the authors used 2D-SIM to show the spatial organisation of alpha-tubulin and CEP44. This improved the resolution compared with other wide-field based imaging. However, to fully dissect the structural organisation of these two large molecules, 3D SIM is necessary: objects that appear to overlap in 2D may in fact be separate in z. Alternatively, at the very minimum, images of the complex in different orientations should be presented.

In figure 4g, the labelling of CEP44 by immunogold staining shows two dots in the representative image. Do the authors think their labelling is incomplete, or do they think the distribution of CEP44 in the centriole does not follow its radial symmetry?

The comparison of secondary structures in figure 4i is not highly informative. How unique is this arrangement of secondary structure elements to MT-binding domains, and how likely is it that the

final tertiary structure is functionally comparable? A multiple-sequence alignment or whole-domain functional prediction might provide further information, should the authors wish to support their argument in this manner.

In figure 7h, the authors present a model of the roles of the proteins investigated in this paper in CCC. Can they comment on how many other proteins could likely be assigned similar roles to CEP44?

In the discussion, the authors make several claims that they do not explain sufficiently.

- What do they mean when they state that the developing centriolar structure acts as a ‘pacemaker’ (line 385) of CCC?

- The authors separately find that CEP295 has a role as a centriole stabilisation factor during SASS6 cartwheel during mitosis, unlike CEP44. They then make the link to *Drosophila* genetics, and use this as an explanation as to why *Drosophila* does not have a CEP44 homologue (lines 376-381). However, this logic is unclear. If CEP295 is required to carry out fewer functions, how does this affect the roles of CEP44, which the authors do not demonstrate interacts directly with CEP295?

15.November.2019

Manuscript NCOMMS-19-20139-T

**Point-to-point responds:**

Reviewer #1 (Remarks to the Author):

In this manuscript, Atorino and colleagues identified a new regulator, CEP44, for the
centriole-to-centrosome conversion (CCC). It has been previously reported that the
occurrence of CCC is dependent on the function of CEP295. The authors found that
CEP44 acts downstream of Cep295 in the process of the CCC, although the physical
interaction between the two proteins is not clear. Depletion of CEP44 in human cells
leads to defects in assembly of PCM and also, to some extent, centriole formation.
The authors provided some evidence that CEP44 forms a complex with POC1B
which is suggested to act as a part of the A-C linker between triplet MTs.
Consistently, CEP44 colocalizes with POC1B at the proximity of the centriole lumen.
According to the analogy of the N-terminal region of CEP44 with EB1 and 3, the
authors generated a presumable MT-binding-deficient mutant of CEP44 and
demonstrated that the mutant did not rescue the phenotype of CEP44 depletion.
Interestingly, the stability assay with cold treatment revealed that CEP295 is more
important to ensure the stability of newly-formed centrioles than CEP44. The authors
further addressed the relationship between structural integrity of centrioles and the
CCC, especially focusing on the arrangement of triplet MTs and the A-C linker. Using
siRNAs against TUBD1 and E1, they show that the structural integrity of new
centrioles is critical for the modification of triplet MTs and the CCC. Overall, the
quality of data is high and convincing, and the manuscript is well written. This
manuscript would provide a new concept that the integrity of centriole wall is

essential for proper CCC. Thus, this study will be of great interest to the centrosome
field and also to cell biologists, and is therefore a strong candidate for publication in
Nature Communications. However, there are several concerns that should be
experimentally addressed in prior to publication.

Main points:

1. In Figure 1, to confirm the defect in the CCC upon CEP44 depletion, it would be
better to distinguish between old and new mother centrioles using markers such as
ODF2 and Cep164. Defects in the CCC should be more frequently detected on new
mother centrioles.

*As suggested by the reviewer 1, we now conducted the analysis to assess whether*
*the CCC defect affects the daughter or the mother centrosome in G1. We depleted*
*CEP44 and stained the new mother centrosome with the marker CEP164. This*
*defined that the defect in CCC affects the CEP164-less centrosome and thus, the*
*daughter one (Supplementary figure 1j-l).*

2. In Figure2, if the authors intend to address the conversion mechanism, the
centriole marker such as centrin, CP110 and CEP97 should be used in this figure. g-
tubulin is not an appropriate marker because its signal is also reduced in the absence
of CEP44.

*As suggested by reviewer 1, a more precise analysis of how CEP44 influences CCC*
*was conducted. In all the immunofluorescence samples, Centrin1 was used as*
*marker to define the position of the centrioles and thus, to assess the loss of both*
*CEP44 upon siCEP295 and of the CCC components upon siCEP44 (Figure 2a-i).*

3. For example, in Figure 2i, this reviewer could not understand why the number of
CEP135 foci was significantly decreased upon CEP44 depletion. Fu et al (NCB,
2015) reported that CEP135 acts upstream of CEP295 in loading this protein to
centrioles. Also, previous studies showed that CEP135 is critical for daughter
centriole formation. As mentioned in the text, CEP135 is not a PCM protein.
Considering the background of CEP135 function, this reviewer is confused with the
result shown in Fig. 2i. The authors should clarify the cause of this phenotype; does
the absence of CEP135 simply reflect a decrease in the number of daughter
centrioles? Otherwise, as the authors claimed, does this phenotype reflect defects in
the CCC although daughter centrioles are somehow formed?

*The relationship between CEP135 and CEP295 seems to be different from organism*
*to organism. In flies (Fu et al. NCB, 2015) CEP295 recruitment to the centriole relies*
*on CEP135, while in human cells this seems to be more reciprocal (Chang et al.*
*JCS, 2016). We therefore determine the temporal recruitment of CEP295 and*
*CEP135 and the role of CEP135 in the CCC. CEP295 is recruited earlier in the cell*
*cycle in RPE1 cells than CEP135 (Supplementary figure 4b-e). Furthermore, upon*
*depletion of CEP135 (Supplementary figure 1a), G1 RPE1 cells showed two*
*centrosomes, which efficiently recruited the PCM component γ -tubulin*
*(Supplementary figure 4f and g).*

4. In Figure 3, the biochemistry testing the physical interaction between CEP44 and
POC1B is rather weak. More fragments of CEP44 should be examined to narrow
down the CEP44 domain responsible for specifically binding to POC1B, but not
POC1A. In addition, to confirm the interaction, in vitro binding assay with the purified

proteins could be useful. If possible, this result will be an important information for
understanding the molecular architecture of basal part of centrioles.

*Thank you very much for this important suggestion. We expanded our analysis of the*
*interaction between CEP44 and POC1B. We purified CEP44-Flag and POC1B-HA*
*recombinant proteins from E.coli and tested the physical interaction of the two*
*proteins in vitro (Figure 3b). Once we confirmed the direct interaction, we narrowed*
*down the domain of CEP44 responsible for the binding to POC1B by generating*
*shorter constructs of CEP44. IB of pull down samples showed that almost the entire*
*CEP44 protein is necessary for the interaction. In fact, besides the full length CEP44,*
*only the construct missing the C-terminal 80 aa was able to bind POC1B. Shorter*
*constructs fail to interact with POC1B (Figure 3e). Moreover, overexpressed full*
*length CEP44 recruited POC1B to cytoplasmic microtubules. Although the N-terminal*
*half of CEP44 bound to microtubules in this experiment, it was unable to recruit*
*POC1B because regions that are critical for the interaction are missing in this CEP44*
*truncation (Supplementary figure 5b-d). These experiments together strongly support*
*our conclusion that CEP44 and POC1B directly interact.*

5. In Figure 4, the phenotype that the CEP44 h5⁻ mutant did not localize to centrioles
is interesting. However, as this mutation seems to be predicted and designed based
on its analogy with EB1 and 3 domains, the ability of h5⁻ mutant protein for binding to
tubulin should be directly tested by biochemistry as in Figure 4h.

*As the reviewer suggested, the ability of the h5⁻ mutant to bind to microtubules in*
*vitro was tested. The h5⁻ mutant was purified from E.coli and subjected to the*
*microtubule-binding assay. The experiment was conducted side by side with the non-*
*mutated CEP44 and confirmed the in vivo behavior of the mutant. The h5⁻ mutant is*

*not able to bind polymerized MTs in comparison to the non-mutated protein (Figure*
*4h).*

If the binding of CEP44 to tubulin is needed, how would Cep295 loss affect the
loading of CEP44 to centrioles? because of defects in the centriole integrity? This
possibility can be tested with siTUBD1 and siTUBE1.

*Considering the centriole wall defect generated from the loss of CEP295 (Figure 6e)*
*and the affinity of CEP44 to MTs (Figure 4h), it is strongly possible that the*
*dependency of CEP44 on CEP295 loading is due to centriole defects. To test*
*whether this is the case, we followed the advise of the reviewer and analyzed the*
*loading of CEP44 to the centrioles upon centriole defects generated by TUBD1 and*
*TUBE1 loss. Upon siTUBD1 and siTUBE1 the loading of CEP44 to the centriole was*
*affected in correlation with the centriole defect, showing that CEP44 localization*
*depends also on the centriole wall integrity (Supplementary Figure 11f and g). In*
*contrast, CEP295 still localized to daughter centrioles upon siTUBD1 and siTUBE1*
*(Supplementary Figure 11d and e).*

Also, did the authors test whether the N-terminal part of CEP44 is sufficient for its
loading to centrioles?

*IF data of overexpressed N-terminal part of CEP44 showed that this construct was*
*sufficient for the loading to centrioles, but was unable to fulfill the full-length protein*
*function. It failed to load POC1B efficiently to centrioles (Supplementary Figure 5k-l*
*and Figure 4j). Furthermore, the mutagenesis of the N-terminal part of CEP44 (h5*
*mutant) disrupts its localization to centrioles (Figure 4j).*

6. In Figure 6, Venoux et al (2013, JCS) reported that POC1A and B act together to
ensure the centriole integrity. Does double-knock-down of both proteins lead to more
significant defects in the centriole structure and CCC? At least, IF-based experiments
testing the loading of CEP295, CEP44, GT335, PCM proteins should be done in this
condition.

*As requested by the reviewer, we assessed the severity of phenotypes generated by*
*the POC1A and POC1B double knockdowns. As Venoux et al. JCS 2013 reported,*
*the double knockdown of POC1A and POC1B affected both the centriole duplication*
*and centriole stability. Also in our hands, G1 cells with POC1A and POC1B double*
*siRNA showed centriole and centrosome loss (Supplementary Figure 6e).*

*In G1 cells that contained both centrioles (judged by centrin1), the recruitment defect*
*of the PCM protein γ -tubulin (Supplementary Figure 6d-f) was in case of the double*
*depletion slightly stronger than in the case of the depletion of the single component*
*POC1B (Figure 3g). Single depletion of POC1A had only a very mild impact*
*(Supplementary Figure 6c for POC1A).*

*We then tested the loading of CEP44 and CEP295 onto centrioles. CEP295 was*
*delocalized in double POC1A+B knockdown but not in the single POC1B depletion*
*(Supplementary figure 6i-k). CEP44 instead was de-localized similarly in double and*
*single POC1B knockdowns (Supplementary figure 6g-h and Figure 3j). These data*
*suggest that there is a redundancy in the function of the POC1A and POC1B*
*proteins, but still significant differences in the function of both proteins.*

7. In Figure 7, this is an excellent experiment addressing the effect of centriole
structure defect solely on the microtubule modification and g-tubulin loading. Using
siTUBD1 and siTUBE1, the loading of CEP295, CEP44 and POC1A,B to centrioles

should be tested. This experiment may address whether the pathway of CEP295-
CEP44-POC1B actually works for the CCC mechanism, or the structural defects of
centrioles by CEP295 depletion just affects the loading of CEP44 and POC1A,B and
other microtubule binders.

*Depletion of TUBD1 and TUBE1 generated loss of structure integrity of centrioles as*
*shown in (Figure 7) and thus defects in CCC. Following the reviewers' suggestion we*
*depleted these two components and tested the loading of CEP295, CEP44 and*
*POC1B. CEP295 localization was not affected by structural defects (Supplementary*
*Figure 11d and e), confirming its role in the early biogenesis of the new daughter*
*centrioles. Differently, CEP44 and POC1B were delocalized upon centriole defects*
*generated by TUBD1 and TUBE1 loss. This suggests that CEP44-POC1B complex*
*localization depends also on the centriole wall integrity (Supplementary Figure 11f-i).*

Minor points:

8. In Figure 3, it would be more informative to indicate the alignment of CEP44 family
proteins in vertebrates to see the evolutionarily conserved and functional domains,
since this protein family is not well characterized thus far.

*The skim of CEP44 protein sequence conservation was added to the Figure 3c. The*
*Supplementary Figure 5a shows the alignment between the protein sequences from*
*vertebrata of the CEP44 conserved domain, annotated as CEP44 domain (see also*
*line 206-207 of the manuscript).*

9. In Figure 5, this reviewer wonders how the cold treatment disrupts unstable or
immature centrioles. Is this because of lack of tubulin modifications on centriolar

microtubules upon CEP295 depletion? But, this might not be the case based on the
result from Fig. 6a-b.

*The loss of glutamylation is for sure not the reason why CEP295 less centrioles are*
*cold sensitive because overexpression of CCP5 did not affect centriole stability*
*(Supplementary figure 10c-d and f-g). Because CEP295 binds more strongly to dCs,*
*we believe that this protein has an additional function not only in CCC but also in*
*centriole stabilization for example by crosslinking tubulin protofilaments in centrioles.*

10. Would the CCC completely depend on the structural integrity of centriole wall?
Otherwise, is there a mechanism separate from it? It would be interesting if the
authors could discuss this issue with their ideas in the revised manuscript.

*In Fig. 7h we discussed that the development of a centriole structure is an important*
*requisites for the recruitment of the PCM proteins but do not exclude any additional*
*mechanism of recruitment of PCM by protein-protein interactions as suggested*
*before (see also lines 484-485 of the manuscript). I have no doubt that CEP295*
*recruits CEP192 as published before. However, somehow this is not working in*
*CEP44 depleted cells, probably because CEP295 is not in state that allows CEP192*
*binding.*

Reviewer #2 (Remarks to the Author):

Centriole-to-centrosome conversion (CCC), which renders daughter centrioles
competent for motherhood, is required for the procentriole to acquire competence for
duplication. While it is well appreciated that centriole maturation and CCC occur
simultaneously, the relationship between the formation of centriole MT triplets and

the recruitment of pericentriolar materials (PCM) remains largely unknown. In this
manuscript, Atorino et al showed that assembly of normal centriole structure is critical
for promoting timely CCC and generating functional centrosomes. Furthermore, they
suggest that Cep44, a component of centriolar lumen, contributes to CCC by
interacting with POC1B and aiding the recruitment of PCM components, such as
Cep152, Cep192, PCNT, etc. Overall, the authors have done a lot of work to
understand how CCC is regulated and how the structure of centriole wall influences
this process. Various knockdown/knockout analyses were carried out to delineate the
CCC pathway. However, the drawback of this study is the lack of understanding at
molecular levels. In addition, the analyses of knockdown cells are not rigorous {there
are no data showing the levels of knockdowns by IB (except Cep44) or intensity
measurements for controls; see below}. Likewise, whether the Cep44-POC1 axis
mediates CCC in a bifurcated or parallel (i.e., independent) pathway remains elusive.
Partially delocalized Cep44 by siCep295 and similarly delocalized Cep152, Cep192,
and Cep135 by siCep44 would make it difficult to convincingly disentangle various
components and their networks that contribute to the CCC pathway.

Major concerns:

1. Fig. 2, Supplementary Fig. 3c-h – As pointed out above, to delineate the Cep295-
mediated CCC pathway, the authors should carry out more rigorous analyses with
proper controls for side-by-side comparison.

*CEP44 and CEP295 depletion efficiencies were assessed upon treatment of the cells*
*with the corresponding siRNA both via IB and IF. Statistical analysis of the depletion*
*efficiencies showed only small variation and thus a strong reproducibility of the*
*depletion of the tested CCC components (Supplementary Figure 1a and i,*

*Supplementary Figure 3e). This was confirmed by the correspondence of the*
*depletion efficiency and the defect generated from it (Supplementary Figure 1i,*
*Supplementary Figure 3e). Using these controls, the analysis of the loss of*
*components showed in Figure 2 and their intensity reductions in Supplementary*
*Figure 3 was carried out. In other case (CEP44 depletion and analysis of the*
*localization of CEP295, CEP44, CEP152, CEP192 and CEP135) the analysis was*
*done side-by-side as suggested by reviewer 2. The same was also the case for*
*TUBD1 and TUBE1 depletions shown in Fig. 7.*

Notably, Cep295 localizes at the periphery of a centriole, whereas Cep44 localizes in
the centriole lumen (as shown in Fig. 4). Thus, as the authors stated in line 347, how
Cep295 can function at the upstream of luminal Cep44 remains a mystery. One
possibility is that Cep295 mediates a bifurcated pathway and one of its branches is
regulated by Cep44. Alternatively, Cep295 and Cep44 may mediate independent
pathways that function in parallel to contribute to CCC.

*To elucidate the missing connection between CEP295 and CEP44, we depleted*
*siTUBD1 and siTUBE1 to generate loss of structure integrity of centrioles as shown*
*in (Figure 7) and thus defects in CCC and tested the loading of CEP295, CEP44 and*
*POC1B. While CEP295 localization was not affected (Supplementary Figure 11d and*
*e) as an early biogenesis factor of the new daughter centrioles, CEP44 and POC1B*
*were delocalized upon centriole defects generated by TUBD1 and TUBE1 loss. This*
*hinted that CEP44-POC1B complex localization depends also on the centriole wall*
*integrity (Supplementary Figure 11f-i), which is also affected upon CEP295 loss*
*(Figure 6a, b and e). Taken together, these data suggest an indirect connection*

*between CEP295 and CEP44, which is based on the formation of a proper centriole*
*wall structure.*
*In addition, we performed co-depletion of CEP295 and CEP44 and then assessed*
*CCC. Co-depletion of CEP295 and CEP44 had the same impact on CCC than single*
*depletion of CEP295 or CEP44. Since the depletion efficiencies were similar in all the*
*set ups, we can conclude that CEP295 and CEP44 function in a linear pathway*
*(Supplementary Figure 3f-h).*

Unfortunately, no IB data are provided except the Cep44 IB shown in Supplementary
Fig. 1a, thus making it difficult to properly interpret the data. Since depletion of one
component may influence the stability of other components in the CCC pathway,
performing IBs for each component in a way that allows cross-examination of all
other components in the pathway would be very helpful.

*As suggested by the reviewer 2, IBs that analyze the depletion efficiencies of the*
*different proteins subject of this study was carried out. As the Supplementary Figure*
*1a shows, depletion of the different proteins was efficiently accomplished upon the*
*treatment with the corresponding siRNA. In addition, we show depletion of CEP295*
*and CEP44 (single and double siRNA) in Supplementary Figure 3h.*

*To have a more accurate depletion efficiency scenario on centrioles, IF of the*
*samples was used in addition to the IB analysis. The IF analysis unveiled that this*
*technique was more precise to determine the depletion efficiency as shown in*
*Supplementary Figure 1 i, Supplementary Figure 3e and Supplementary Figure 5h.*

In addition, determining the severity of siCep295 siCep44 double knockdowns in
comparison to siCep295 or siCep44 alone will help propose whether the pathway is

bifurcated or is composed of two independent pathways functioning in parallel to
regulate the CCC. In the case of parallel pathways, Cep44 does not function at the
downstream of Cep295.

*To test this interesting point raised by reviewer 2, we depleted CEP44 and CEP295*
*both separately and together. Because the depletion of the single siRNA or the*
*double knockdown did not generate a more severe phenotype concerning the CCC*
*defect (Supplementary figure 3f-h), despite similar depletion efficiencies*
*(Supplementary Fig. 3h), we concluded that CEP44 follows CEP295 in a linear*
*pathway, functioning at the downstream of CEP295.*

Furthermore, the intensity measurements shown in Supplementary Fig. 3 lack
important controls. For instance, for Supplementary Fig. 3d, the authors should
provide the level of Cep295 intensity depleted by siCEP295 under the same
conditions. This may allow one to assess whether ~50% reduction in Cep44 signal
intensities achieved by siCEP295 is meaningful. If Cep295 depletion were as
complete as for Cep44, then the ~50% reduced Cep44 signal could be resulted from
gross structural defects associated with Cep295 depletion. Given the essential role of
Cep295 in organizing a functional centrosome (Tsuchiya Y, et al, Nat Comm, 2016;
Fu, J, et al, NCB, 2016), this point is especially important.

*Thank you very much for raising this point. As pointed out above, CEP44 and*
*CEP295 depletion efficiencies were assessed upon treatment of the cells with the*
*corresponding siRNA both via IB and IF.*

*What we show is:*

*CEP44 depletion affects 65% CEP44 but not CEP295 (Figure 1e and Figure 2b)*
*CEP295 depletion affects CEP44 by 84% (Figure 2d).*

*Cep295 depletion affects CEP295 by 87% (Supplementary figure 3e).*

*CEP44 depletion affects CEP152, CEP192, CEP135, POC1B but not CEP295.*

*From these data it is clear that CEP44 functions downstream of CEP295.*

Likewise, quantified Cep44 signal intensities for both siControl and siCep44 cells
must be shown side-by-side to relatively assess the significance of ~50% reduced
signal intensities for Cep152, Cep192, and Cep135 shown in Supplementary Fig. 3f-
305 h.

*We have performed these controls. The depletion efficiency of CEP44 in comparison*
*to the siRNA control is shown in Supplementary Figure 1a by IB. We further show*
*depletion efficiency of CEP44 by IF in Figure 1e. Using the same cells (side-by-side)*
*we have tested localization of CEP295, CP152, CEP192 and CEP135 upon siCEP44*
*and siControl (Figure 2). The experiment shows that while CEP295 localization is not*
*affected by CEP44 depletion, CEP152 (70%), CEP292 (70%) and CEP135 (75%)*
*localization with centrioles is strongly affected. These data show that CEP44*
*functions downstream of CEP295. In addition, CEP152, CEP135 and CEP192*
*function downstream of CEP44.*

At present, it is premature to suggest the importance of the Cep44-POC1B axis in
regulating downstream PCM proteins, such as Cep152, Cep193, and PCNT.

*We have deepened our analysis on the interaction of CEP44 and POC1B. In*
*particular, we show that both proteins directly interact. This conclusion is based on in*
*vitro binding experiments with purified proteins (Figure 3b) and on the observation*
*that overexpressed CEP44 that binds to cytoplasmic microtubules is able to recruit*
*POC1B to this localization.*

*We show in a new side-by-side experiment that the impact of CEP44 depletion on*
*POC1B is stronger than the other way round (Figure 3i-j and Supplementary figure*
*5j). This suggests that POC1B functions downstream of CEP44 despite this*
*interdependency (indicated by the double arrows in Figures 3k and 7h).*
*The N-terminal CEP44 fragment that binds to microtubules but not POC1B when*
*overexpressed partially suppresses the CCC defect of CEP44 depletion (Figure 4k:*
*from 40% empty plasmid control to 67% in the NT-CEP44 overexpression). POC1B*
*recruitment in CEP44 depleted cells rises from 40% in empty plasmid control to 55%*
*in the NT- CEP44 sample (Supplementary figure 5l). This experiment allow us to*
*conclude that CEP44 has functions independent of the POC1B binding site probably*
*through the stabilization of centriole microtubules. It also suggests that this CEP44*
*function is already sufficient to recruit some POC1B even when the POC1B binding*
*site in CEP44 is missing. However, it is also clear that recruitment of POC1B via*
*CEP44 is necessary to achieve full CCC. In the revised manuscript, we now discuss*
*this “complex” relationship between CEP44 and POC1B (see also lines 426-442 in*
*the manuscript).*

2. Fig. 3 – the authors show that Cep44 interacts with POC1B. However, with the
data provided in Fig. 3b, apparently carried out using recombinant proteins as affinity
ligands, it is difficult to judge how efficiently they interact with each other. If it is co-IP
analysis, then input % needs to be shown.

*As suggested by the reviewer 2, a deeper analysis of the interaction between CEP44*
*and POC1B was conducted. We purified CEP44-Flag and POC1B-HA recombinant*
*proteins from E.coli and show physical interaction of the two proteins in vitro (Figure*
*3b). In Figure 3a (Figure 3b of the previous version of the Figure file), the CEP44 pull*

*down experiment, we added the information of % of the input and eluate of the pull*
*down experiment performed with CEP44-Flag recombinant protein. Finally, using an*
*IP approach, we have mapped the critical region in CEP44 for the interaction with*
*POC1B and show that a truncated form of CEP44 (NT-CEP44, Figure 3e) binds to*
*microtubules (Supplementary figure 7f) but is unable to interact with POC1B (Fig.*
*3e). Interestingly, this mutant form of CEP44 (NT CEP44) when overexpressed*
*partially suppresses the CCC defect of CEP44 depletion as discussed above (Figure*
*4k) and now also discussed in the manuscript.*

The lack of sufficient colocalization shown in Figs. 4d and 4f strongly suggests that
these two proteins may not form a stable complex. Therefore, their partially
interdependent colocalization shown in Fig. 4e-h could be due to a structural defect
in centrioles by Cep44 RNAi. To properly assess the data, the authors should provide
either IBs or quantified signal intensities for control and RNAi cells.

*CEP44 and POC1B perfectly co-localize in procentrioles. Later in the cell cycle after*
*centriole elongation, both proteins show partial co-localization.*

*We have added additional data to the manuscript that support the notion that CEP44*
*and POC1B interact directly. As discussed above and in the manuscript, while it is*
*clear from these collective evidences that CEP44 and POC1B interact, CEP44 has*
*functions in the centriole that are independent of its ability to interact with POC1B (as*
*now shown by Figure 4k). We have modified the discussion in order to contribute to*
*these new findings (see also lines 426-442 of the manuscript).*

3. Fig. 4 – The authors nicely showed that the Cep44-h5 mutant's defect in MT
binding cripples the CCC. In the light of one of their major findings that Cep44

interacts with POC1B to regulate the CCC (Fig. 3), generating a Cep44 mutant
defective in POC1B binding could be more meaning for this work.

*Thank you very much this suggestion. We were able to separate the contributions of*
*CEP44 to the microtubule binding activity and POC1B recruitment. First, CEP44 h5⁻*
*mutant, which lacks the ability to bind MTs (Figure 4h), is unable to rescue the CCC*
*defect upon siCEP44 (Figure 4j and k) and it delocalizes POC1B from the*
*centrosomes if overexpressed (Supplementary Figure 8d and e). Consistent with*
*these properties, CEP44 h5⁻ shows a dominant negative CCC phenotype.*
*Second, the NT-CEP44 construct is not able to bind POC1B but still binds to*
*microtubules (Figure 3e, Supplementary Figure 7f). This mutant was partially able to*
*rescue the CCC defect of CEP44 depletion (Figure 4j and k) and to localize POC1B*
*to centrioles in siCEP44 depleted cells (Supplementary Figure 5k and l). This*
*indicates that CEP44 provides functions even when it does not interact with POC1B.*
*We discuss this finding though in lines 426-442 of the manuscript.*

Interestingly, unlike the MT plus-end binding EB1, Cep44 localizes to the centriole
lumen under physiological conditions. Therefore, the authors may explore whether
the capacity of Cep44 to interact with the lumen-localizing POC1B helps target
Cep44 to this location. This notion can be discussed in the Discussion section.

*As nicely noticed by the reviewer, the localization of CEP44 partially depends on the*
*localization of the lumen protein POC1B (Figure 4h and j). This indicates that POC1B*
*has some impact on the recruitment or binding efficiency of CEP44 to centrioles.*

*However, it is also clear that NT-CEP44 that lacks the POC1B interaction region*
*associates with centrioles.*

*Figures 3k and 7h depict these experiment findings by the double arrow between*
*CEP44 and POC1B. In addition, we now discuss this complex relationship between*
*CEP44 and POC1B in the manuscript.*

4. fig. 6 – IB or quantified signal intensities showing the levels of knockdowns should
be provided for better assessment of the data. If the levels of knockdowns are similar
between Cep44 and POC1B, then these two proteins may not be in the linear
pathway, as proposed in Figs. 3i and 7h. Rather, the delocalization of POC1B in
Cep44 RNAi cells could be due to an indirect consequence of Cep44 RNAi-induced
structural defects.

*As suggested by reviewer 2, CEP295, CEP44 and POC1B depletion efficiencies*
*were now assessed both via IB and IF and the strong reproducibility of the depletion*
*via the more accurate technique (IF) was used as internal control (Supplementary*
*Figure 1i, Supplementary Figure 3e, Supplementary Figure 5h).*

*As discussed above and in lines of the discussion, the relationship between CEP44*
*and POC1B has a number of interesting facets. Our data suggest that full CCC*
*requires the CEP44-POC1B interaction. However, it is also clear that CEP44 can*
*impact on CCC without its POC1B binding site (Figure 4k).*

5. Supplementary Fig. 4g – Again, the normalized level of Cep44 signal intensities in
siCep44 cells is necessary to comparatively assess the significance of the diminished
POC1B signals in siCep44 cells. If the Cep44 depletion is near complete, as shown
in Supplementary Fig. 1a, then ~50% reduction in the POC1B signal may suggest
that the Cep44-POC1B interaction in Fig. 3 is less likely significant.

*In Figure 3h-j and Supplementary figure 5j we have performed a side-by-side*
*experiment analyzing the impact of CEP44 and POC1B depletions on the localization*
*of CEP44 and POC1B. CEP44 depletion affects CEP44 by 62% and POC1B by 60%.*
*POC1B depletion affects POC1B by 54% and CEP44 by 19%. This suggests that*
*CEP44 has a stronger impact on POC1B localization on centrioles than POC1B on*
*CEP44. This uneven interdependency between both proteins is reflected by the*
*uneven arrows in Figures 3k and 7h.*

6. Supplementary Fig. 6 – The data show that Cep295 preferentially localizes to
daughter centrioles and functions as dC stabilizing factor. The author should examine
whether this is the case for Cep44 and POC1, as suggested in their model.

*We appreciate this comment. However, we do not suggest in our model that CEP44*
*and POC1B preferentially bind to the dC. Figure 4d shows that CEP44 is of equal*
*intensity at the mC and dC. In contrast, daughter centrioles have less POC1B than*
*mother centrioles (Figure 4d).*

Minor comments:

1. Fig. 4 – Schematic diagrams for the localized Cep44 and POC1B signals in 4d and
Cep295 and tubulin signals at the daughter centriole in 4f will be helpful

*We followed the suggestion of the reviewer 2 and added schematic representation of*
*the localization of CEP44 and POC1B and CEP295 to the Figure 4d and f, on the*
*side of the 2D-SIM images.*

2. Supplementary Fig. 1 – ~30-60% defect in the recruitment of PCNT and gamma-
tubulin after a near-complete depletion of Cep44 suggests that Cep44 controls only a
part of the CCC pathway.

*Comparison between the depletion efficiency of CEP44 (Figure 1e) and the defect in*
*recruitment of γ -tubulin (Figure 1f) or PCNT (Supplementary Figure 1h) show a*
*correspondence between the depletion of CEP44 and the phenotype as shown for*
*example in the Supplementary Figure 1i.*

3. Line 800 – the distance of “C- and B-tubule” should be changed to “B- and C-
tubule”.

*As suggested, the segment “C- and B-tubule” was changed to “B- and C-tubule”.*

4. Line 91 – eliminate “)” from “---Cenp-F))”.

*As suggested, “)” was eliminated from “---Cenp-F))”.*

5. Line 135 – downstream “of” CEP295

*As suggested, “downstream CEP295” was corrected with “downstream of CEP295”.*

Reviewer #3

(Remarks to the Author): Review paper General comments: The authors attempt
to demonstrate that successful centriole-to-centrosome conversion (CCC) relies on
perfect structural integrity of the centriole. The authors focus their study almost
exclusively on the study of the essential, but otherwise uncharacterised, protein
CEP44 and its interaction with other CCC proteins. Why do the authors focus on this
particular protein in the context of CCC integrity?

*Our interest in characterizing CEP44 out of novel centrosomal proteins raised on the*
*essentiality of CEP44 gene in human cells as described in the first paragraph of the*
*results section (see line 76-78 of the manuscript).*

Since there is a multitude of proteins involved in the CCC process, their choice of
CEP44 should be much more clearly justified. The authors report a number of
interesting individual findings, based on good experiments and mostly sound
interpretation. However, the paper is poorly structured and the logic is not very well
presented. We feel the paper should be presented differently (the title is far too
general). It should be clear that this is a study on CEP44's potential role in CCC
primarily, not on the centriole structural integrity's importance for CCC. It would be
more appropriate to argue that centriole structural integrity offers a potential
explanation for observed CEP44 phenotypes.

*Based on this comment, we followed the advise of the reviewer and changed the title*
*from "The formation of bona fide centriole wall is necessary for the centriole-to-*
*centrosome conversion" to "CEP44 ensures the formation of bona fide centriole wall,*
*a prerogative for the centriole-to-centrosome conversion".*

We would suggest the paper be structured along these lines (figure 5 does not fit in
well with the rest of the paper and should not be presented here):

*As suggested by the Editor, Figure 5 was not removed from the structure of the*
*paper. We also consider this data set as very important because it describes a novel*
*function of CEP295 in centriole biogenesis.*

*Base on the comments of the reviewer we optimized the logical flow of the paper. In*
*particular, we explained better why we have done certain experiments. The flow is:*

*CEP44 is an essential component of centrioles involved in CCC. The function of*
*CEP44 is downstream of CEP295 (Figures 1 and 2). We then ask the question how*
*CEP44 executes this function: in complex with CEP295 or in association with an*
*additional factor? These experiments identified POC1B as CEP44 interactor while*
*CEP295 did not interact with CEP44. Because CEP295 is on the outer wall of*
*centrioles and POC1B on the inner wall, this raised the question of the localization of*
*CEP44 (Figure 4). CEP295 was described as a factor that stabilizes centrioles. Is this*
*common to CEP44 and POC1B? (Figure 5). How do CEP295, CEP44 and POC1B*
*promote CCC considering their distinct localization on centrioles (Figure 6). We*
*finally test our model by the depletion of TUBD1 and TUBE1 (Figure 7).*

Introduction

- • There are gaps in our understanding of CCC, which may be explained by the
- involvement of uncharacterised proteins.
- • One such uncharacterised protein is CEP44, which has been suggested to be a
- centriolar protein in a published screen.
- o Why CEP44, out of this list?

*We structured the Introduction as suggested by reviewer 3.*

Results

- • Figure 1: CEP44 is indeed a centriolar protein, and it is essential for CCC
- • Next question: what is its role in CCC?
- • Figure 2: CEP44 influences the recruitment of proteins downstream of CEP295, but
- not of CEP295 itself.

*This is the flow of our paper.*

o Rephrase the figure title! Influencing a downstream pathway is not the same thing
as being downstream in a pathway.

*We rephrased the figure title as suggested by reviewer 3.*

• Next question: is it a component of this pathway (downstream)?

• Supplementary table 1: CEP44 interactor analysis shows only POC1A and POC1B
(?), not CEP295

o Were these really the only hits?

*POC1B and POC1A were the only hits that we found in the CEP44-Flag pull-down*
*samples by mass-spectrometry.*

*In BioGrid and IntAct the CEP44-POC1B interaction is also reported. CEP295 was*
*not reported as an interactor of CEP44 in BioGrid and IntAct.*

*We preferred to have Figure 3 before Figure 4 in order to introduce the interaction*
*between CEP44 and POC1B prior to the deeper analysis of the localization of the*
*proteins. In this way, we can compare a side-by-side localization analysis of CEP295,*
*CEP44 and POC1B (Figure 4a-f).*

• Figure 4: evidence of different localisation -> need to mention this here

*We show the different localizations of CEP44-POC1B and CEP295 (lines 254- 271 of*
*the manuscript) on the daughter and mother centrioles.*

o (Note this needs to be addressed more fully in the discussion, particularly in the
last paragraph)

*As suggested by the reviewer, the localization of CEP44 and POC1B during centriole*
*biogenesis is now discussed in lines 429-433 of the manuscript.*

• Next question: if CEP44 is not involved in the CEP295 pathway, does it interact
with other characterised centriolar proteins/pathways? Considering its localisation to
the centriole lumen, which proteins are attractive candidates and what does this
suggest regarding potential functions (in structure)?

*We follow this logic and performed a pull-down/mass spectrometry screen for CEP44*
*interactors that identified POC1A and B but not CEP295. Figure 3 then confirms that*
*POC1B is a direct CEP44 interactor with a function in CCC, while CEP295 did not*
*bind to CEP44.*

• Figure 3: CEP44 interacts with POC1B, and this complex is needed for CCC

• Next question: what is the role of the complex? Considering POC1B has a role in
centriole maintenance, is it structural?

*We have addressed the role of the complex in Figure 4j and k. This result is*
*discussed on lines 426-442.*

o The leap between this complex and the investigation of the role of centriole
structural stability in CCC needs to be made clear and explicit.

*The role of POC1B in the CCC was investigated based on its interaction with CEP44*
*and based on its role in centriole stabilization as previously published by Venoux et*
*al. JCS, 2013.*

• Figure 6: conversion molecules are needed for structural integrity, including CEP44
and POC1B

• Next question: can this role account for the phenotype? Is it a potential explanation
of the effect of CEP44 depletion on CCC?

*This is the flow of the manuscript.*

• Figure 7: comprising centriole structural integrity via interference with tubulin epsilon
and delta phenocopies CEP44 depletion

*This is our argumentation.*

Conclusion

• The effect of CEP44 depletion observed in figure 1 may be due to it compromising
centriole structural integrity.

• CEP295 cannot recruitment its downstream proteins if this structure is
compromised.

• Therefore, CCC can be compromised (in disease) by loss of function of more
proteins than just those involved in the key CCC pathway downstream of CEP295.

*These points are outlined in the discussion.*

Specific comments: The figures are not consistent in their lay-out. For instance, in
figure 2, the colour scheme of the merged images is inconsistent, with the nucleus
(DAPI stain) only being blue in some.

*The layout of the Figure 2 was changed based on the suggestion of reviewer 3. All*

*the images in that figure follow the same color scheme.*

Much of the figures' content is presentation of single representative images. This is
sometimes accompanied by quantification of a larger dataset, but this is missing for
the intensity profiles presented in figure 4. How reproducible are these graphs?

*The intensity profiles in the Figure 4 are related to the images next to it. CEP295*
*localization was already published. These published data are consistent with Figure*
*4c and f. Additional examples of the SIM localization of CEP44 and POC1B are now*
*shown in Supplementary figure 7a-d. In addition, CEP44 SIM the localization*
*corresponds with the immuno-EM in Figure 4g and Supplementary Figure 7e. In*
*summary, the graphs are indeed reproducible.*

In figure 4, the authors used 2D-SIM to show the spatial organisation of alpha-tubulin
and CEP44. This improved the resolution compared with other wide-field based
imaging. However, to fully dissect the structural organisation of these two large
molecules, 3D SIM is necessary: objects that appear to overlap in 2D may in fact be
separate in z. Alternatively, at the very minimum, images of the complex in different
orientations should be presented.

*Following the suggestion of the reviewer, more 2D-SIM images of the complex*
*CEP44-POC1B localization were added to the Supplementary Figure 7b.*

In figure 4g, the labelling of CEP44 by immunogold staining shows two dots in the
representative image. Do the authors think their labelling is incomplete, or do they
think the distribution of CEP44 in the centriole does not follow its radial symmetry?

*Due to the concern of the poor immunogold labeling of CEP44, the image in Figure*
*4g was changed with a clearer example of the CEP44 immunogold labeling. The*
*previous image was moved to the Supplementary Figure 7e and in addition a further*
*example was added to Supplementary Figure 7e. These three images show that the*
*labeling of CEP44 follows the radial symmetry of centrioles.*

*However, labeling efficiency is with 2-6 gold particles per centriole cross-section*
*moderate. However, this is a common phenomenon of post-labeling immuno-EM*
*because only the surface exposed antigens are accessible to the antibodies.*

The comparison of secondary structures in figure 4i is not highly informative. How
unique is this arrangement of secondary structure elements to MT-binding domains,
and how likely is it that the final tertiary structure is functionally comparable?

*There is no published data showing a muster of secondary structure organization*
*and/or folding properties of MT-binding domains. Our conclusions are based on the*
*strong similarity of the secondary structure organization of CEP44 protein sequence*
*to the characterized MT-binding proteins EB1 and EB3. Interestingly, an in silico*
*modeling (SwissMODEL from the Exspasy platform) of CEP44 MT-binding domain,*
*based on the sequence similarity (18%) with the protein IFT81, shows a similar*
*tertiary structure organization as the one of the crystallized EB1 and EB3 MT-binding*
*domains.*

A multiple-sequence alignment or whole-domain functional prediction might provide
further information, should the authors wish to support their argument in this manner.

*Following the suggestion of the reviewer 3, in Supplementary Figure 5a the N-*
*terminal conserved domain of the CEP44 protein sequence alignment from*
*vertebrata is shown. The N-terminus of CEP44 was annotated as CEP44 domain*
*(see also line 206-207 of the manuscript). At the end, this sequence analysis did not*
*give us additional information.*

In figure 7h, the authors present a model of the roles of the proteins investigated in
this paper in CCC. Can they comment on how many other proteins could likely be
assigned similar roles to CEP44?

*Considering our finding that structural defects impair CCC, it is likely that depletion of*
*most proteins with a function in centriole biogenesis have a similar phenotype.*

In the discussion, the authors make several claims that they do not explain
sufficiently.

• What do they mean when they state that the developing centriolar structure acts as
a 'pacemaker' (line 385) of CCC?

*The section mentioned by the reviewer was changed to "This finding suggests a new*
*function for the developing centriolar structure as one of the most important*
*requisites for recruitment of PCM proteins". The meaning of the "pacemaker" concept*
*was substituted with the idea of "requisite".*

• The authors separately find that CEP295 has a role as a centriole stabilisation
factor during SASS6 cartwheel during mitosis, unlike CEP44. They then make the
link to Drosophila genetics, and use this as an explanation as to why Drosophila does
not have a CEP44 homologue (lines 376-381). However, this logic is unclear. If

CEP295 is required to carry out fewer functions, how does this affect the roles of
CEP44, which the authors do not demonstrate interacts directly with CEP295?
*Because of the reviewers' concern about the logic of the comparison to CCC in flies,*
*we revised the discussion (see lines 471-478 of the manuscript).*

Reviewers' comments:

Reviewer #1 (Remarks to the Author):

Atorino and colleagues made a lot of efforts to improve the quality of the data showing the molecular mechanisms of the CCC with a particular focus on the Cep295-Cep44-POC1B axis. They addressed all of my concerns by performing new experiments and modification of the manuscript. Overall, it seems to this reviewer that the current version of the manuscript is now ready for publication in Nature Communications.

Reviewer #2 (Remarks to the Author):

The revised version is much improved, providing recommended controls required for better interpreting the data. However, resolving a few issues below will be very helpful to strengthen this work.

1. While I generally agree with the authors' main conclusion of the work that constructing a normal centriole wall is critical for the centriole-to-centrosome conversion (CCC), I am not sure whether the CEP44-POC1B interaction that they demonstrate in this study is significant.

2. That being said, detailed experimental procedures should be provided for the Fig. 3a, b, e and Fig. 4h. The authors stated that Fig. 3a is anti-Flag pulldown from RPE cells expressing CEP44-Flag. By reading the rebuttal letter, it looks like the Fig. 3b was done with recombinant proteins purified from *E. coli* (this was not mentioned in the main text). Fig. 3e appears to be carried out using a method similar to Fig. 3a. Since the binding efficiency appears to be very low in all cases, the authors should provide detailed buffer conditions, etc in a separate method section. Also, it would be much more informative, if a silver gel (if not Coomassie-stained gel) could be provided for the Fig. 3b (and Fig. 4h below). Immunoblotting analysis is an odd way of detecting the ligand and bound targets for in vitro binding assays carried out with purified proteins. There is no way to tell the purity of the proteins used for the binding analyses. A silver gel may allow one to estimate binding efficiency and/or stoichiometry, etc. The same is true for Fig. 4h.

3. In an extension of the #2 comment above, it is not clear whether the CEP44-POC1B interaction is significant. The authors showed that NT-Flag, which fails to bind to POC1B (Fig. 3e), still exhibits a significant level of activity to recruit POC1B to centriole and partially rescues the CCC (Supplementary Fig. 5k-l). These observations suggest that the CEP44-POC1B pathway is likely not a linear pathway in vivo. The authors may need to have a branching arrow from CEP44?

4. Supplementary Fig. 5j – Should “>2” be “<2”?

5. Supplementary Fig. 5 (l) is not described in the legend. Should it be Supplementary Fig. 5 (k-l)?

Reviewer #3 (Remarks to the Author):

The structure of the paper has improved substantially, and that most of my concerns have been addressed.

I still have the following remaining comments:

* I think the conclusion that structure influences protein-protein interactions in and of itself is already a well-established concept, though it is interesting that even small differences can have dramatic effects in this particular case. It would be of added value in the ‘discussion’ section if the authors could comment on how likely these minor defects are to arise in early stages of and therefore be causative of particular diseases?

* In the changed title (line 484), the authors use the term ‘prerogative’. The definition of ‘prerogative’ is “a right or privilege exclusive to a particular individual or class”, i.e., it is predominantly a political term. We would suggest ‘requirement’ as an alternative.

* There are still some inconsistencies in the colour scheme of the figures. For instance, in figure 5c, only one nucleus in the merged images is blue, and in supplementary figure 1, not all nuclei in the merged images are blue.

* Minor comments:

- There are still some typos in the manuscript (e.g., ‘engage’ in line 52, ‘arching’ in line 449, ‘less’ rather than ‘fewer’ in line 120, many unhyphenated adjectives).

- Line 61 (and 410): 'human cells' is not very specific (and the authors alternate between 'flies' and *D. melanogaster*).

18.12.2019

Manuscript NCOMMS-19-20139-T

**Point-to-point responds:**

Reviewer #1 (Remarks to the Author):

Atorino and colleagues made a lot of efforts to improve the quality of the data
showing the molecular mechanisms of the CCC with a particular focus on the
Cep295-Cep44-POC1B axis. They addressed all of my concerns by performing new
experiments and modification of the manuscript. Overall, it seems to this reviewer
that the current version of the manuscript is now ready for publication in Nature
Communications.

*We thank the Reviewer #1 for the positive comments on the re-submitted work.*

Reviewer #2 (Remarks to the Author):

The revised version is much improved, providing recommended controls required for
better interpreting the data. However, resolving a few issues below will be very
helpful to strengthen this work.

1. While I generally agree with the authors' main conclusion of the work that
constructing a normal centriole wall is critical for the centriole-to-centrosome
conversion (CCC), I am not sure whether the CEP44-POC1B interaction that they
demonstrate in this study is significant.

*In vivo (Figure 3a and 3e: CEP44-Flag IP experiment, Supplementary fig. 5d-f:*
*CEP44-dependent targeting) and in vitro (Figure 3b and Supplementary fig. 5b and c)*
*experiments, together with dC specific CEP44-POC1B co-localization data, provided*

*strong evidences of CEP44 binding to POC1B. This binding may be transient – we*
*do not claim that CEP44 and POC1B form a stable complex.*
*It is also clear from our data that the CEP44-POC1B relationship is more complex*
*than the simple formation of a stable complex. The CEP44 NT lacking the POC1B*
*interaction domain already plays a role in CCC (around 15-25% in the rescue*
*experiments of Figure 4k and Supplementary fig. 6b). However, the presence of the*
*full-length CEP44 is needed to recruit POC1B and fully rescue the CCC phenotype*
*(60%, Figure 4k and Supplementary fig. 6b and see correlation of POC1B loss upon*
*siCEP44 in Supplementary fig. 5l). We describe this relationship between CEP44 and*
*POC1B in the Discussion, line 435 onwards: “However, further studies indicated that*
*both proteins showed a more complex localization interdependency indicating*
*additional principles for POC1B centriole location than only binding to CEP44.”*
*We also propose in the Discussion (lines 446-448) that the CEP44-POC1B*
*interaction is transient: “Based on these findings, we propose that transient CEP44-*
*POC1B complex formation is needed early in centriole biogenesis to create a*
*centriole structure that then allows CEP44-independent recruitment of POC1B.”. This*
*remark was added because of the concern of reviewer 2.*

2. That being said, detailed experimental procedures should be provided for the Fig.
3a, b, e and Fig. 4h. The authors stated that Fig. 3a is anti-Flag pulldown from RPE
cells expressing CEP44-Flag. By reading the rebuttal letter, it looks like the Fig. 3b
was done with recombinant proteins purified from E. coli (this was not mentioned in
the main text). Fig. 3e appears to be carried out using a method similar to Fig. 3a.
Since the binding efficiency appears to be very low in all cases, the authors should
provide detailed buffer conditions, etc in a separate method section.

*We apologize for not being too clear about the experimental designs. Figure 3a is a*
*Flag IP experiment. This is now clearly indicated in the result and legend. Fig. 3b is*
*an in vitro binding experiment with purified, recombinant proteins. This is stated on*
*line 200-202: “The interaction between CEP44 and POC1B was further confirmed*
*using E.coli purified recombinant proteins (Fig. 3b, immunoblot and Supplementary*
*Fig. 5b and c, Coomassie).” Fig. 3b is an immunoblot. As requested by reviewer 3,*
*we have added a Coomassie Blue stained gel of the purified proteins to*
*Supplementary Figure 5c. The in vitro binding experiment is also shown as*
*Coomassie Blue stained gel in Supplementary Figure 5b. Figure 3e is a CEP44-Flag*
*IP experiment similar to Figure 3a. In Figure 3e different CEP44 constructs were*
*expressed in RPE1 cells, followed by anti-Flag IP and analysis of the samples by IB.*
*This is now clearly stated in the figure legend.*
*To have a better understanding of the experimental procedures behind the*
*experiment in Fig. 3a, b, e and Fig.4h, the details of the experimental procedures and*
*buffer conditions were added to the manuscript METHODS section. See paragraphs*
***“Protein purification”, “Microtubule binding assay”, “CEP44-Flag IP” and “CEP44-***
***POC1B in vitro binding assay”.***

Also, it would be much more informative, if a silver gel (if not Coomassie-stained gel)
could be provided for the Fig. 3b (and Fig. 4h below). Immunoblotting analysis is an
odd way of detecting the ligand and bound targets for in vitro binding assays carried
out with purified proteins. There is no way to tell the purity of the proteins used for the
binding analyses. A silver gel may allow one to estimate binding efficiency and/or
stoichiometry, etc.

*As suggested from the Reviewer #2, Coomassie-stained gels were added to the*
*Supplementary figures. In detail, Supplementary fig. 5b is a Coomassie Blue-stained*
*gel showing the samples used in the IB of Figure 3b. Supplementary fig. 5c is a*
*Coomassie Blue-stained gel of CEP44-Flag and POC1B-HA purified recombinant*
*proteins used in the Figure 3b ran separately in two different lines to show the purity*
*grade of the single proteins.*

*The necessity to blot the samples of the experiment in Figure 3b arose to confirm*
*that the most represented band of the elution sample (beside CEP44-Flag) shown in*
*Coomassie-Blue stained gel was POC1B-HA.*

The same is true for Fig. 4h.

*Supplementary fig. 8f is a Coomassie Blue stained gel in which the recombinant*
*proteins α -tubulin, GST-Flag, CEP44-Flag and CEP44-Flag h5 were ran separately*
*to show the purity of the single proteins used in the MT-binding assay of Figure 4h.*

*The necessity to blot the samples of the experiment in Figure 4h arose from the fact*
*that the CEP44-Flag and tubulin run in SDS-PAGE gels with the same mobility as*
*can be seen in Supplementary Figure 8f. Thus, without the IB, we just would detect*
*one protein band in the Coomassie Blue stained gel (sum of tubulin and CEP44-*
*Flag). The experiment would be inclusive.*

3. In an extension of the #2 comment above, it is not clear whether the CEP44-
POC1B interaction is significant. The authors showed that NT-Flag, which fails to
bind to POC1B (Fig. 3e), still exhibits a significant level of activity to recruit POC1B to
centriole and partially rescues the CCC (Supplementary Fig. 5k-l). These
observations suggest that the CEP44-POC1B pathway is likely not a linear pathway

in vivo. The authors may need to have a branching arrow from CEP44?

*It is true that CEP44 NT half is already able to partially rescue both POC1B*
*recruitment (15% rescue, Supplementary fig. 6b) and the CCC (25% rescue, Figure*
*4k) and we stated a caveat in the “DISCUSSION” section of the manuscript. But it is*
*also clear that the binding of CEP44 to POC1B, which is generated from the full-*
*length protein (Figure 3e), is necessary to rescue the remnant 35% of CCC defect*
*and 40% of POC1B localization. As mentioned above, we do not claim that the*
*CEP44-POC1B interaction forms and then is stable. Instead we believe that a*
*transient interaction between CEP44 and POC1B helps to recruit POC1B to*
*centrioles early in dC formation (as discussed on lines 446-448). This conclusion*
*reflects the interaction of both proteins as shown by IP, in vitro binding, localization*
*dependency and the overlap in the localization of both proteins early in dC formation.*
*As suggested from the Reviewer #2, a branching arrow was added to the linear*
*pathway showed in the model in Figure 7h to underline the role of CEP44 NT itself in*
*the CCC.*

4. Supplementary Fig. 5j – Should “>2” be “<2”?

*Thanks to the Reviewer #2 we changed the “>2” into “<2”. This detail is of a big*
*significance for the conclusions of the work.*

5. Supplementary Fig. 5 (l) is not described in the legend. Should it be

Supplementary Fig. 5 (k-l)?

*The description of Supplementary Fig. 5l was added in the Supplementary legends*
*file. Now it is described in Supplementary fig. 6a and b.*

Reviewer #3 (Remarks to the Author):

The structure of the paper has improved substantially, and that most of my concerns
have been addressed.

I still have the following remaining comments:

* I think the conclusion that structure influences protein-protein interactions in and of
itself is already a well-established concept, though it is interesting that even small
differences can have dramatic effects in this particular case. It would be of added
value in the 'discussion' section if the authors could comment on how likely these
minor defects are to arise in early stages of and therefore be causative of particular
diseases?

*In agreement with the Editor comment, we did not speculate in discussion about*
*potential disease relevance. It will be for sure an important question to answer in*
*future projects.*

* In the changed title (line 484), the authors use the term 'prerogative'. The definition
of 'prerogative' is "a right or privilege exclusive to a particular individual or class", i.e.,
it is predominantly a political term. We would suggest 'requirement' as an alternative.

*As suggested by the Reviewer #3, the authors agreed to change the word*
*"prerogative" with "requirement" in the manuscript title, being the term "prerogative"*
*predominantly a political term.*

* There are still some inconsistencies in the colour scheme of the figures. For
instance, in figure 5c, only one nucleus in the merged images is blue, and in

supplementary figure 1, not all nuclei in the merged images are blue.

*To generate figures with consistent in the colour scheme, Figure 5c and*

*Supplementary fig. 10c colours were changed matching the all-over colour scheme of*

*the Figures. Supplementary fig. 1a colour scheme was not changed to better*

*appreciate the localization of the different markers at the centrosome.*

* Minor comments:

- There are still some typos in the manuscript (e.g., 'engage' in line 52, 'arching' in

line 449, 'less' rather than 'fewer' in line 120, many unhyphenated adjectives).

*The typos "engage" was changed to "engaged", "arching" to "arcing" and "less" to*

*"fewer". Furthermore, many un-hyphenated adjectives were found and hyphenated.*

- Line 61 (and 410): 'human cells' is not very specific (and the authors alternate

between 'flies' and D. melanogaster').

*In both line 61 and 410 (now 416) the definition of "human cells" was changed to*

*"human cultured cell lines" as different cell lines were used for the centrosome*

*maturation studies. Moreover, al the terms "flies" were changed to D. melanogaster,*

*being under the flies the organism used to study centrosomes.*

REVIEWERS' COMMENTS:

Reviewer #2 (Remarks to the Author):

This version is much improved, essentially eliminating the concerns that I have had about the CEP44-POC1B interaction. One thing that I'd like to suggest is that the authors should consider placing the Supplementary Fig. 5b in the main text (or swap it with the immunoblotted Fig. 3b). The in vitro binding shown in Supplementary Fig. 5b suggests that the interaction between CEP44 and POC1B is very strong, exhibiting an approximately 1:2 binding stoichiometry (an eyeball estimate!). We cannot learn much from the Fig. 3b except that the proteins in the Supplementary Fig. 5b are indeed right proteins.

Note a typo in line 929—change “CEP44-Fag” to “CEP44-Flag”.

21.01.2020

*Manuscript NCOMMS-19-20139-T*

***Point-to-point responds:***

Reviewer #2 (Remarks to the Author):

This version is much improved, essentially eliminating the concerns that I have had
about the CEP44-POC1B interaction. One thing that I'd like to suggest is that the
authors should consider placing the Supplementary Fig. 5b in the main text (or swap
it with the immunoblotted Fig. 3b). The in vitro binding shown in Supplementary Fig.
5b suggests that the interaction between CEP44 and POC1B is very strong,
exhibiting an approximately 1:2 binding stoichiometry (an eyeball estimate!). We
cannot learn much from the Fig. 3b except that the proteins in the Supplementary
Fig. 5b are indeed right proteins.

*Based on the Reviewer suggestion, the Figure 3b was swapped with Supplementary*
*Figure 5b.*

Note a typo in line 929—change “CEP44-Fag” to “CEP44-Flag”.

*The expression “CEP44-Fag” was changed to “CEP44-Flag” in the indicated text*
*position.*
